# Increasing adult-born neurons protects mice from epilepsy

Swati Jain[1], John J LaFrancois[1], Kasey Gerencer[1†], Justin J Botterill[2], Meghan Kennedy[1], Chiara Criscuolo[1,3], Helen E Scharfman[1,3,4]*

[1]Center for Dementia Research, The Nathan S. Kline Institute for Psychiatric Research, Orangeburg, United States; [2]Department of Anatomy, Physiology, & Pharmacology, College of Medicine, University of Saskatchewan, Saskatoon, Canada; [3]Departments of Child and Adolescent Psychiatry, New York University Grossman School of Medicine, New York, United States; [4]Departments of Neuroscience & Physiology, Psychiatry, and the Neuroscience Institute, New York University Grossman School of Medicine, New York, United States

## eLife assessment

In this manuscript, Jain and colleagues explore whether increasing adult-born neurons is protective against status epilepticus and the development of spontaneous recurrent seizures (chronic epilepsy) in a mouse pilocarpine model of temporal lobe epilepsy. This is an **important** work that provides **solid** data, contradicting previous studies on suppressing chronic seizures by reduction in adult-born neurons.

*For correspondence:
Helen.Scharfman@nki.rfmh.org

Present address: †Department of Psychology, The University of Maine, Orono, United States

**Abstract** Neurogenesis occurs in the adult brain in the hippocampal dentate gyrus, an area that contains neurons which are vulnerable to insults and injury, such as severe seizures. Previous studies showed that increasing adult neurogenesis reduced neuronal damage after these seizures. Because the damage typically is followed by chronic life-long seizures (epilepsy), we asked if increasing adult-born neurons would prevent epilepsy. Adult-born neurons were selectively increased by deleting the pro-apoptotic gene *Bax* from Nestin-expressing progenitors. Tamoxifen was administered at 6 weeks of age to conditionally delete *Bax* in Nestin-CreER^T2^*Bax*^fl/fl^ mice. Six weeks after tamoxifen administration, severe seizures (status epilepticus; SE) were induced by injection of the convulsant pilocarpine. After mice developed epilepsy, seizure frequency was quantified for 3 weeks. Mice with increased adult-born neurons exhibited fewer chronic seizures. Postictal depression was reduced also. These results were primarily in female mice, possibly because they were more affected by *Bax* deletion than males, consistent with sex differences in *Bax*. The female mice with enhanced adult-born neurons also showed less neuronal loss of hilar mossy cells and hilar somatostatin-expressing neurons than wild-type females or males, which is notable because loss of these two hilar cell types is implicated in epileptogenesis. The results suggest that selective *Bax* deletion to increase adult-born neurons can reduce experimental epilepsy, and the effect shows a striking sex difference. The results are surprising in light of past studies showing that suppressing adult-born neurons can also reduce chronic seizures.

## Introduction

It has been shown that neurogenesis occurs in the hippocampal dentate gyrus (DG) during the adult life of mammals (*Taupin, 2006*; *Gage et al., 2008*; *Altman, 2011*; *Kempermann, 2012*; *Kazanis, 2013*). It is important to note that this idea was challenged recently (*Paredes et al., 2018*; *Sorrells*

*et al., 2018*) but afterward more studies provided support for the original idea (*Boldrini et al., 2018*; *Kempermann et al., 2018*; *Tartt et al., 2018*; *Moreno-Jiménez et al., 2019*; *Tobin et al., 2019*).

In the DG, adult-born neurons are born in the subgranular zone (SGZ; *Altman and Das, 1965*; *Kaplan and Hinds, 1977*; *Altman, 2011*). Upon maturation, newborn neurons migrate to the granule cell layer (GCL; *Cameron et al., 1993*), develop almost exclusively into GCs, and integrate into the DG circuitry like other GCs (*Ramirez-Amaya et al., 2006*; *Kempermann et al., 2015*).

Prior studies suggest that the immature adult-born GCs can inhibit the other GCs (*Ash et al., 2023*) especially when they are up to 6-weeks-old (*Drew et al., 2016*). By inhibition of the GC population, young adult-born GCs could support DG functions that require GCs to restrict action potential (AP) discharge, such as pattern separation (*Sahay et al., 2011a*; *Sahay et al., 2011b*). Indeed suppressing adult neurogenesis in mice appears to weaken pattern separation (*Clelland et al., 2009*; *Nakashiba et al., 2012*; *Niibori et al., 2012*; *Tronel et al., 2012*) and increasing adult neurogenesis improves it (*Sahay et al., 2011b*).

In addition, inhibition of the GC population by young adult-born GCs could limit excessive excitation from glutamatergic input and protect the cells in the DG hilus and hippocampus that are vulnerable to excitotoxicity. Thus, strong excitation of GCs can cause excitotoxicity of hilar neurons, area CA1 pyramidal cells, and area CA3 pyramidal cells (*Scharfman and Schwartzkroin, 1990a*; *Scharfman and Schwartzkroin, 1990b*; *Sloviter, 1994*; *Scharfman, 1999*; *Sloviter et al., 2003*). Indeed, increasing adult-born neurons protects hilar neurons, and CA3 from neuronal loss 3 days after severe seizures are induced by the convulsant pilocarpine (*Jain et al., 2019*).

The seizures induced by kainic acid or pilocarpine are severe, continuous, and last several hours, a condition called *status epilepticus* (SE). The neuronal injury in the hippocampus after SE has been suggested to be important because it is typically followed by chronic seizures (epilepsy) in rodents and humans, and has been suggested to cause the epilepsy (*Falconer et al., 1964*; *Sloviter, 1994*; *Cavalheiro et al., 1996*; *Herman, 2002*; *Mathern et al., 2008*; *Dudek and Staley, 2012*; *Dingledine et al., 2014*). Chronic seizures involve the temporal lobe, so the type of epilepsy is called temporal lobe epilepsy (TLE). In the current study, we asked if increasing adult -born neurons can protect from chronic seizures in an animal model of TLE. We used a very common method to induce a TLE-like syndrome, which involves an injection of the muscarinic cholinergic agonist pilocarpine at a dose that elicits SE. Several weeks later, spontaneous intermittent seizures begin and continue for the lifespan (*Scorza et al., 2009*; *Botterill et al., 2019*; *Lévesque et al., 2021*; *Whitebirch et al., 2022*). Seizure frequency, duration, and severity were measured by continuous video-EEG with 4 electrodes to monitor the hippocampus and cortex bilaterally.

It is known that SE increases adult neurogenesis (*Parent and Kron, 2012*). SE triggers a proliferation of progenitors in the week after SE (*Parent et al., 1997*). Although many GCs that are born in the days after SE die in subsequent weeks by apoptosis, some survive. Young neurons that arise after SE and migrate into the GCL may suppress seizures by supporting the inhibition of GCs because adult-born GCs in the normal brain inhibit GCs when they are young (*Drew et al., 2016*; *Ash et al., 2023*). In addition, after SE, the newborn GCs in the GCL can exhibit low excitability (*Jakubs et al., 2006*). However, some neurons born after SE mismigrate to ectopic locations such as the hilus (hilar ectopic GCs), where they can contribute to recurrent excitatory circuits that promote seizures (*Scharfman et al., 2000*; *Parent and Lowenstein, 2002*; *Scharfman, 2004*; *Scharfman and Hen, 2007b*; *Parent and Murphy, 2008*; *Scharfman and McCloskey, 2009*; *Zhan et al., 2010*; *Myers et al., 2013*; *Cho et al., 2015*; *Althaus et al., 2019*; *Zhou et al., 2019*). Since the hilar ectopic GCs are potential contributors to epileptogenesis, we also studied whether enhancing adult-born neurons would alter the number of hilar ectopic GCs.

Mossy cells are a major subset of glutamatergic hilar neurons which are vulnerable to excitotoxicity after SE (*Scharfman, 1999*; *Sloviter et al., 2003*). During SE, mossy cells may contribute to the activity that ultimately leads to widespread neuronal loss (*Botterill et al., 2019*). However, surviving mossy cells can be beneficial after SE because they inhibit spontaneous chronic seizures in mice (*Bui et al., 2018*). Another large subset of vulnerable hilar neurons co-express GABA and somatostatin (SOM; *Sloviter, 1987*; *de Lanerolle et al., 1989*; *Freund et al., 1992*; *Sun et al., 2007*) and correspond to so-called HIPP cells (neurons with **hi**lar cell bodies and axons that project to the terminal zone of the **p**erforant **p**ath; *Han et al., 1993*). HIPP cells are important because they normally inhibit

GCs and have the potential to prevent seizures. Therefore, we studied mossy cells and SOM cells in the current study.

The results showed that increasing adult-born neurons protects mossy cells and hilar SOM cells and reduces chronic seizures. Remarkably, the preservation of hilar mossy cells and SOM cells, and the reduction in chronic seizures, was found in females primarily. The sex difference may have been due to a greater ability to increase adult-born neurons in females than males, consistent with sex differences in *Bax*- and caspase-dependent cell death (*Forger et al., 2004*; *Siegel and McCullough, 2011*).

The results are surprising because prior studies that suppressed neurogenesis reduced chronic seizures. Therefore, taken together with the results presented here, both increasing and suppressing adult-born neurons appear to reduce chronic seizures. How could this be? Past studies suggested that suppressing adult-born neurons led to a reduction in chronic seizures because there were fewer hilar ectopic granule cells. In the current study, increasing adult-born neurons may have reduced chronic seizures for another reason. Regardless, the present data suggest a novel and surprising series of findings which, taken together with past studies, suggest that adult-born neurons can be targeted in multiple ways to reduce chronic seizures in epilepsy.

## Results
## Increasing adult-born neurons reduced the duration of pilocarpine-induced SE

### General approach
The first experiment addressed the effect of increasing adult-born neurons on pilocarpine-induced SE in Nestin-CreER$^{T2}$*Bax*$^{fl/fl}$ mice (called 'Cre+,' below). To produce Nestin-CreER$^{T2}$*Bax*$^{fl/fl}$ mice, hemizygous Nestin-CreER$^{T2}$ mice were bred with homozygous *Bax*$^{fl/fl}$ mice. Littermates of Cre+ mice that lacked Cre (called 'Cre-,' below) were also treated with tamoxifen and were controls.

*Figure 1A1* shows the experimental timeline. Tamoxifen was injected s.c. once per day for 5 days to delete *Bax* from Nestin-expressing progenitors. After 6 weeks, a time sufficient for a substantial increase in adult-born neurons (*Drew et al., 2016*; *Jain et al., 2019*), pilocarpine was injected s.c. to induce SE.

*Figure 1A2* shows the experimental timeline during the day of pilocarpine injection. The location of electrodes for EEG are shown in *Figure 1B*. Mice monitored with EEG were implanted with electrodes 3 weeks before SE (see Methods). Examples of the EEG are shown in *Figure 1C* for Cre- and Cre+ mice and details are shown in *Figure 1—figure supplement 1*.

### Effects of increasing adult-born neurons on SE
The latency to the first seizure after pilocarpine injection was measured for all mice (with and without EEG electrodes; *Figure 1D*) or just those that had EEG electrodes (*Figure 1E*). When mice with and without electrodes were pooled, the latency to the onset of first seizure was similar in both genotypes (Cre-: 47.2±4.8 min, n=27; Cre+: 45.3±3.9 min, n=28; Student's t-test, t(53)=0.3, p=0.761; *Figure 1D1*).

The total number of seizures was quantified until 2 hr after pilocarpine injection because at that time diazepam was administered to decrease the severity of SE. The total number of seizures were similar in both genotypes (Cre-: 3.0±0.2 seizures; Cre+: 2.8±0.1 seizures; Student's t-test, t(54)=0.9, p=0.377; *Figure 1D2*).

Interestingly, when the sexes were separated, Cre+ females had a shorter latency to the first seizure than all other groups (*Figure 1D3*). Thus, a two-way ANOVA with genotype (Cre- and Cre+) and sex (female and male) as main factors showed a main effect of sex (F(1,51)=4.31; p=0.043) with Cre+ females exhibiting a shorter latency compared to Cre+ males (Cre+ females, 34.3±3.4 min, n=15; Cre+ males, 57.9±5.6 min, n=13; Tukey's post-hoc test, p=0.026) but not other groups (Cre- female, 46.7±9.3 min, n=12; Cre- males, 47.4±4.4 min, n=15; all p>0.344; *Figure 1D3*). There was no effect of genotype (F(1,49)=0.75; p=0.305) or sex (F(1,49)=0.62; p=0.436) on the total number of seizures by two-way ANOVA (*Figure 1D4*).

When adult neurogenesis was suppressed by thymidine kinase activation in GFAP-expressing progenitors, the severity of the first seizure was worse, meaning it was often convulsive rather than non-convulsive (*Iyengar et al., 2015*). Therefore, we examined the severity of the first seizure. These

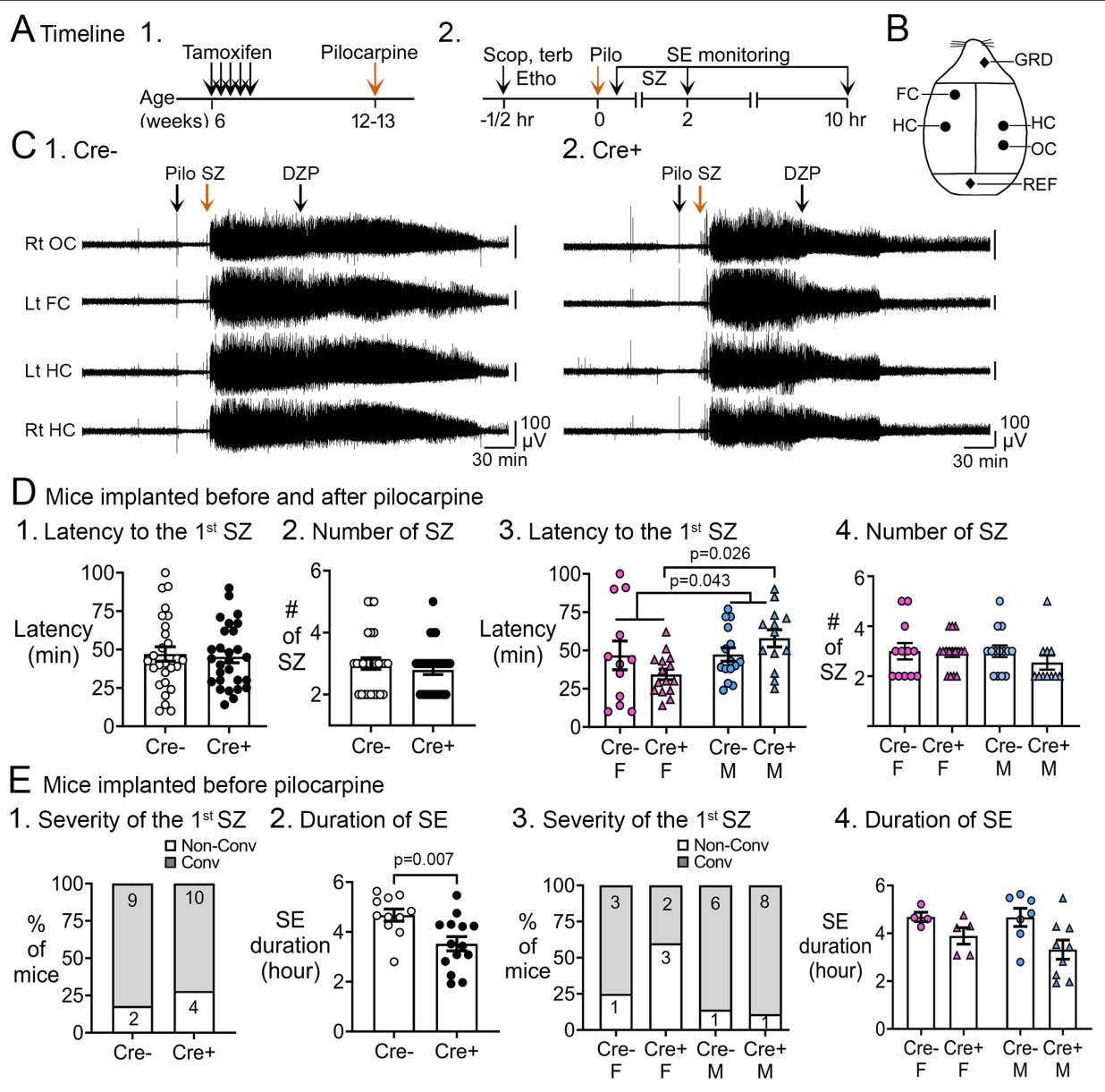

**Figure 1.** Pilocarpine-induced SE in Cre+ and Cre- mice. (**A**) The experimental timeline is shown. (1) Tamoxifen was injected 1 /day for 5 days in 6-week-old Nestin-CreER$^{T2}$*Bax*$^{fl/fl}$ mice. Six weeks after the last tamoxifen injection, mice were injected with pilocarpine (Pilo) at a dose that induces SE. (2) On the day of the pilocarpine injection, one group of mice without electrodes for electroencephalographic (EEG) was monitored for behavioral seizures for 2 hr after the pilocarpine injection. Another group of mice were implanted with EEG electrodes 3 weeks prior to pilocarpine injection. In these mice, video and EEG was used to monitor SE for 10 hr after pilocarpine injection. (**B**) Locations to implant EEG electrodes are shown. Four circles represent recording sites: left frontal cortex (Lt FC), left hippocampus (Lt HC), right hippocampus (Rt HC), and right occipital cortex (Rt OC). Two diamonds represent ground (GRD), and reference (REF) electrodes. (**D**) Pooled data for mice that were implanted with EEG electrodes and unimplanted mice. These data showed no significant genotypic differences but there was a sex difference. (1) The latency to the onset of the first seizure was similar in both genotypes (t-test, p=0.761). The seizure was a behavioral seizure >stage 3 of the Racine scale (unilateral forelimb jerking). For this figure and all others, detailed statistics are in the Results. (2) The number of seizures in the first 2 hr after pilocarpine injection was similar in both genotypes (t-test, p=0.377). (3) After separating males and females, females showed a shorter latency to the onset of the first seizure compared to males (two-way ANOVA, p=0.043); Cre+ females had a shorter latency to the first seizure relative to Cre+ males (Tukey's post-hoc test, p=0.026). (4) The number of seizures in the first 2 hr after pilocarpine injection were similar in males and females (two-way ANOVA, p=0.436). (**E**) Implanted mice. These data showed a significant protection of Cre+ mice on SE duration. (1) The severity of the first seizure (non-convulsive or convulsive) was similar between genotypes (Fisher's exact test, p=0.093). (2) Cre+ mice had a shorter duration of SE than Cre- mice (t-test, p=0.007). (3) After separating males and females, the first seizure was mostly non-convulsive in Cre+ females compared to Cre- females (60% vs 14%) but no groups were statistically different (Fisher's exact tests, p>0.05). (4)

*Figure 1 continued on next page*

*Figure 1 continued*

Once the sexes were separated, there was no effect of sex by two-way ANOVA but there was a trend in Cre+ males to have a shorter SE duration than Cre- males (Tukey's post-hoc test, p=0.078).

The online version of this article includes the following figure supplement(s) for figure 1:

**Figure supplement 1.** Examples of the electroencephalogram (EEG) during SE.

**Figure supplement 2.** Incidence of SE in the unimplanted and implanted mice.

**Figure supplement 3.** Power during SE in Cre+ female and Cre- female mice.

**Figure supplement 4.** The latency to SE, interval between SE and diazepam administration, and interval between pilocarpine and diazepam injections were not significantly different in experimental groups.

analyses were conducted only with mice implanted with electrodes because only with the EEG can one determine if a seizure is non-convulsive. A non-convulsive seizure was defined as an EEG seizure without movement. When sexes were pooled, the proportion of mice with a non-convulsive first seizure was not different (Cre-: 18.2%, 2/11 mice; Cre+: 28.6%, 4/14 mice Fisher's exact test, p>0.999; *Figure 1E1*). However, when sexes were separated, the first seizure was non-convulsive in 60% of Cre+ females (3/5 mice) whereas only 25% of Cre- females had a first seizure that was nonconvulsive (1/4 mice), 14% of Cre- males (1/7 mice), and 11% of Cre+ males (1/9 mice; *Figure 1E3*). Although the percentages suggest differences, i.e., Cre+ females were protected from an initial severe seizure, the differences were not significantly different (Fisher's exact test, p=0.166; *Figure 1E3*).

The duration of SE was shorter in Cre+ mice compared to Cre- mice (Cre-: 280.5±14.6 min, n=11; Cre+: 211.4±17.2 min, n=14; Student's t-test, t(23)=0.30, p=0.007; *Figure 1E2*). When the sexes were separated, the effects of genotype were modest. A two-way ANOVA showed that the duration of SE was significantly affected by genotype (F(1,21)=6.7; p=0.017) but not sex (F(1,21)=5.04; p=0.487). Cre+ males showed a trend for a shorter SE duration compared to Cre- males (Cre- males: 280.1±22.8 min, n=7; Cre+ males: 199.1±24.1 min, n=9; Tukey's post-hoc test, p=0.078; *Figure 1E4*). Cre+ females had a mean SE duration that was shorter than Cre- females, but it was not a significant difference (Cre- females: 281.0±11.8 min, n=4; Cre+ females: 233.4±20.5 min, n=5; p=0.485; *Figure 1E4*). More females would have been useful, but the incidence of SE in females was only 42.8% if they were implanted with EEG electrodes (*Figure 1—figure supplement 2*). In contrast, the incidence of SE in the unimplanted females was 100%, a significant difference by Fisher's exact test (p<0.0001; *Figure 1—figure supplement 2*). In males, the incidence of SE was also significantly different in implanted and unimplanted mice (implanted males, 70.4%; unimplanted males, 100%; Fisher's exact test, p<0.0001, *Figure 1—figure supplement 2*).

## Power

We also investigated power during SE (*Figure 1—figure supplement 3*). The baseline was measured, and then power was assessed for 5 hr, after SE had ended. Power was assessed in 20-min consecutive bins. Females were used in this analysis (Cre- and Cre+). Two-way RMANOVA with genotype and time as main factors showed no effect of genotype for any frequency range: delta (1–4 Hz, F(1,7)=1.61; p=0.245); theta (4–8 Hz, F(1,7)=1.75; p=0.227); beta (8–30 Hz, F(1,7)=1.65; p=0.240); low gamma (80 Hz, F(1,7)=0.29; p=0.174); high gamma (80–100 Hz, F(1,7)=0.17; p=0.689). There was a significant effect of time for all bands (delta, p=0.003; theta, p=0.002, beta, low gamma, and high gamma, p<0.001), which is consistent with the declining power in SE with time.

## Role of diazepam

Diazepam was administered earlier in females during SE than in males, and this could have influenced the results. However, the timing of SE was not significantly different in females than in males (*Figure 1—figure supplement 4*). Also, diazepam was administered the same way in all Cre+ and Cre- females similarly but only the Cre+ females were protected as discussed below.

In summary, Cre+ mice did not show extensive differences in SE except for SE duration, which was shorter.

## Increasing adult-born neurons decreased chronic seizures

### Numbers and frequency of chronic seizures

Continuous video-EEG was recorded for 3 weeks to capture chronic seizures (*Figure 2A*). Representative examples of chronic seizures are presented in *Figure 2B*. All chronic seizures were convulsive. First, we analyzed data with sexes pooled (*Figure 2C*) and the total number of chronic seizures were similar in the two genotypes (Cre-: 22.6±3.0 seizures, n=18; Cre+: 21.3±1.6 seizures, n=17; Student's t-test, t(33)=0.15, p=0.882; *Figure 2C1*). The frequency of chronic seizures were also similar among genotypes (Cre-: 1.1±0.14 seizures/day, n=18; Cre+: 1.0±0.08 seizures/day, n=17; Welch's t-test, t(26)=0.37, p=0.717; *Figure 2D1*).

Data were then segregated based on sex and a two-way ANOVA was conducted with genotype and sex as main factors. There was a main effect of genotype (F(1,32)=4.26; p=0.047) and sex (F(1,32)=12.46; p=0.001) on the total number of chronic seizures and a significant interaction between sex and genotype (F(1,32)=8.54; p=0.006). Tukey's post-hoc tests showed that Cre+ females had ~half the chronic seizures of Cre- females (Cre- female: 44.6±10.2 seizures, n=7; Cre+ female: 22.6±2.0 seizures, n=9; p=0.004; *Figure 2C*). However, Cre+ males and Cre- males had a similar number of chronic seizures (Cre- male: 16.1±1.6 seizures, n=12; Cre+ male: 19.9±2.7 seizures, n=8; p>0.999; *Figure 2C2*). Cre- females also had more seizures than Cre- males (p<0.001) and Cre+ males (p=0.005; *Figure 2C2*).

Results for seizure frequency were similar to results comparing total numbers of seizures. There was a main effect of genotype (F(1,32) 4.18; p=0.049) and sex (F(1, 32)=11.96; p=0.002) on chronic seizure frequency, and a significant interaction between sex and genotype (F(1,32)=8.29; p=0.007). Cre+ female mice had approximately half the seizures per day as Cre- females (Tukey's post-hoc test, Cre- female: 2.1±0.5 seizures/day; Cre+ female: 1.1±0.1 seizures/day; p=0.004; *Figure 2D2*). Cre- females also had more seizures than Cre- males (p<0.001) and Cre+ males (p=0.005; *Figure 2D2*).

### Additional analyses

While reviewing the data for each mouse plotted in *Figure 2C2 and D*, one point appeared spurious in the Cre-females, potentially influencing the comparison. The seizures in this mouse were more than 2 x the standard deviation of the mean. Although not an outlier using the ROUT method (see Methods), we were curious if removing the data of this mouse would lead to a difference in the statistical results. There was still a main effect of sex (F(1,31)=16.04; p=0.0004) with a significant interaction between sex and genotype (F(1,31)=9.20; p=0.005) and Cre+ females had significantly fewer seizures than Cre- female mice (p=0.020; *Figure 2—figure supplement 1A1*). Tests for seizure frequency led to the same conclusions (*Figure 2—figure supplement 1A2*). These data suggest that spurious data point was not the reason for the results.

All mice were included in the analyses above, both those implanted and unimplanted during SE. Mice which were unimplanted prior to SE were implanted at approximately 2–3 weeks after pilocarpine to study chronic seizures. Because implantation affected the incidence of SE (discussed above), we asked if chronic seizures were different in implanted and unimplanted mice. The total number of chronic seizures (F(1,17)=1.33, p=0.265) and seizure frequency (F(1,17)=1.27, p=0.276) were similar, suggesting that implantation did not influence chronic seizures (*Figure 2—figure supplement 1B*).

There were no significant differences in mortality associated with SE or chronic seizures. For quantification, we examined mortality during SE and the subsequent 3 days, 3 days until the end of the 3-week-long-EEG recording period, or both (*Figure 2—figure supplement 2A*). Graphs of mouse numbers (*Figure 2—figure supplement 2B1*) or percentages of mice (*Figure 2—figure supplement 2B2*) were similar: groups (Cre- females, Cre+ females, Cre- males, Cre+ males) were not significantly different (Fisher's exact tests, all p >0.05).

### Mean duration of individual chronic seizures

To evaluate the duration of individual seizures at the time mice were epileptic, two measurements were made. First, the durations of each seizure of a given mouse were averaged, and then the averages for Cre- mice were compared to the averages for Cre+ mice (*Figure 2E1*). There was no difference in the genotypes (Cre-: 46.8±2.9 s, n=17; Cre+: 43.4±2.5 s, n=16; Student's t-test, t(31)=0.89, p=0.379; *Figure 2E1*).

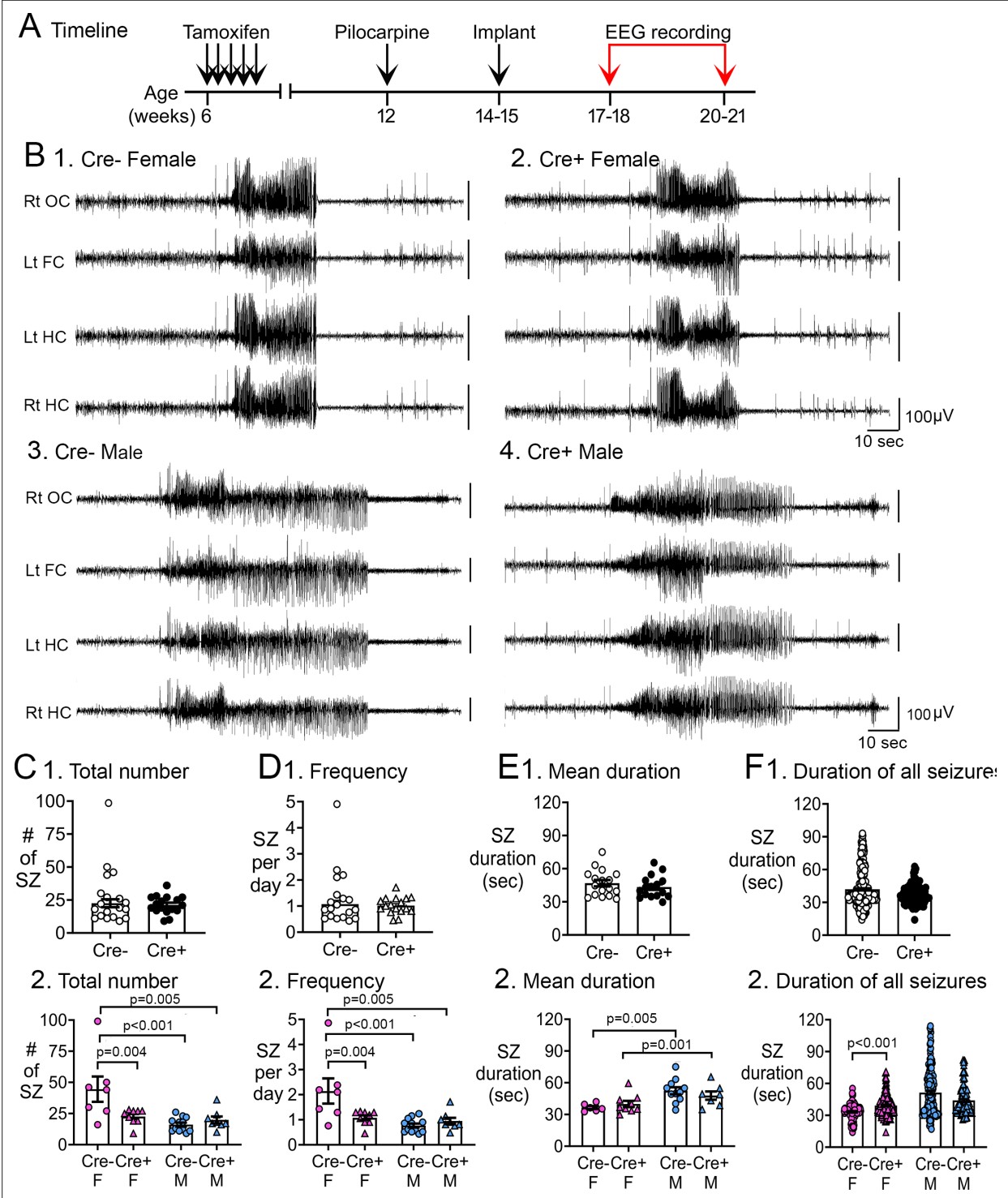

**Figure 2.** Reduced chronic seizures in Cre+ mice. (**A**) The experimental timeline is shown. Six weeks after pilocarpine injection, continuous video-EEG was recorded for 3 weeks to capture chronic seizures. Mice that were unimplanted prior to SE were implanted at 2–3 weeks after pilocarpine injection. (**B**) Representative examples of 2 min-long EEG segments show a seizure in a Cre- (1, 3) and Cre+ (2, 4) mouse. (**C**) Numbers of chronic seizures. (1) Pooled data of females and males showed no significant effect of genotype on chronic seizure number. The total number of seizures during 3 weeks of recording were similar between genotypes (t-test, p=0. 882). (2) After separating data based on sex, Cre+ females had fewer seizures than Cre- females (two-way ANOVA followed by Tukey's post-hoc test, p=0.004). There was a sex difference in control mice, with fewer seizures in Cre- males compared to Cre- females (p<0.001). Cre- females also had more seizures than Cre+ males (p=0.005). (**D**) Chronic seizure frequency. (1) Pooled data of females and males showed no significant effect of genotype on chronic seizure frequency. The frequency of chronic seizures (number of seizures per day) were

*Figure 2 continued on next page*

*Figure 2 continued*

similar (Welch's t-test, p=0.717). (2) Seizure frequency was reduced in Cre+ females compared to Cre- females (Tukey's post-hoc test, p=0.004). There was a sex difference in control mice, with higher seizure frequency in Cre- females compared to Cre- males (p<0.001) and Cre+ males (p=0.005). (**E**) Seizure duration per mouse. (1) Each data point is the mean seizure duration for a mouse. Pooled data of females and males showed no significant effect of genotype on seizure duration (t-test, p=0.379). (2) There was a sex difference in seizure duration, with Cre- males having longer seizures than Cre- females (Tukey's post-hoc test, p=0.005). Because females exhibited more postictal depression (see *Figure 3*), corresponding to spreading depolarization (*Ssentongo et al., 2017*), the shorter female seizures may have been due to truncation of seizures by spreading depolarization. (**F**) Seizure durations for all seizures. (1) Every seizure is shown as a data point. The durations were similar for each genotype (Mann-Whitney *U* test, p=0.079). (2) Cre+ females showed longer seizures than Cre- females (two-way ANOVA followed by Tukey's post-hoc test, p<0.001). Cre+ females may have had longer seizures because they were protected from spreading depolarization.

The online version of this article includes the following figure supplement(s) for figure 2:

**Figure supplement 1.** Additional analyses of chronic seizures.

**Figure supplement 2.** Mortality was not significantly affected by genotype or sex.

When separated by sex, a two-way ANOVA showed that female seizure durations were shorter than males (F(1,29)=12.42; p=0.001). However, this was a sex difference, not an effect of genotype (F(1,29)=0.033; p=0.856; *Figure 2E2*), with Cre- female seizure duration shorter than Cre- male seizure duration (Tukey's post-hoc test, p=0.005), and the same for Cre+ females compared to Cre+males (p=0.001; *Figure 2E2*). One reason for the sex difference could be related to the greater incidence of postictal depression in females (see below), because that suggests spreading depolarizations truncated the seizures in females but not males.

The second method to compare seizure durations compared the duration of every seizure of every Cre- and Cre+ mouse. In the previous comparison (*Figure 2E1*), every mouse was a data point, whereas here every seizure was a data point (*Figure 2F1*). The data were similar between genotypes (Cre-: 41.9±0.9 s; Cre+: 36.8±0.7 s; Mann-Whitney *U* test, *U* statistic, 18873, p=0.079; *Figure 2F1*). When the sexes were separated, a Kruskal-Wallis test was significant (Kruskal-Wallis statistic, 69.30, p<0.001). Post-hoc tests showed that Cre+ females had longer seizure durations than Cre- females (Cre- female: 33.2±0.7 s; Cre+ female: 39.3±0.6 s, p<0.001; *Figure 2F2*). Cre+ females may have had longer seizures because they were protected from spreading depolarizations that truncated seizures in Cre- females. Seizure durations were not significantly different in males (Cre- male: 51.4±1.7 s; Cre+ male: 44.0±1.3 s, p=0.298; *Figure 2F2*).

## Postictal depression

Postictal depression is a debilitating condition in humans where individuals suffer fatigue, confusion, and cognitive impairment after a seizure. In the EEG, it is exhibited by a decrease in the EEG amplitude immediately after a seizure ends relative to baseline. In recent years the advent of DC amplifiers made it possible to show that postictal depression is often associated with spreading depolarization (*Ssentongo et al., 2017*), a large depolarization shift that is accompanied by depolarization block. As action potentials are blocked there are large decreases in input resistance leading to cessation of synaptic responses. As ion pumps are activated to restore equilibrium, there is recovery and the EEG returns to normal (*Somjen, 2001*; *Hartings et al., 2017*; *Herreras and Makarova, 2020*; *Lu and Scharfman, 2021*).

We found that males had little evidence of postictal depression but it was common in females (*Figure 3*), a sex difference that is consistent with greater spreading depolarization in females (*Eikermann-Haerter et al., 2009*; *Bolay et al., 2011*; *Kudo et al., 2023*). As shown in *Figure 3A1*, a male and a female showed a robust spontaneous seizure (selected from the 3-week-long recording period when mice are epileptic). However, the end of the seizure in the male did not exhibit a decrease in the amplitude of the EEG (relative to baseline) but it did in the female (*Figure 3A1-3*). In *Figure 3A2-3*, the male seizure is on the left and the female seizure is on the right. For quantification (*Figure 3B-C*), the mean peak-to-trough amplitude of the EEG 25–30 s before the seizure was compared to the mean amplitude of the EEG during the maximal depression of the EEG after the seizure. If the depression was more than half, the animal was said to have had postictal depression.

When all chronic seizures were analyzed (n=274), the number and percentages of seizures with postictal depression were reduced in Cre+ females compared to Cre- females (Fisher's exact test, p=0.009; *Figure 3B*). There was a sex difference, with all females showing more postictal depression

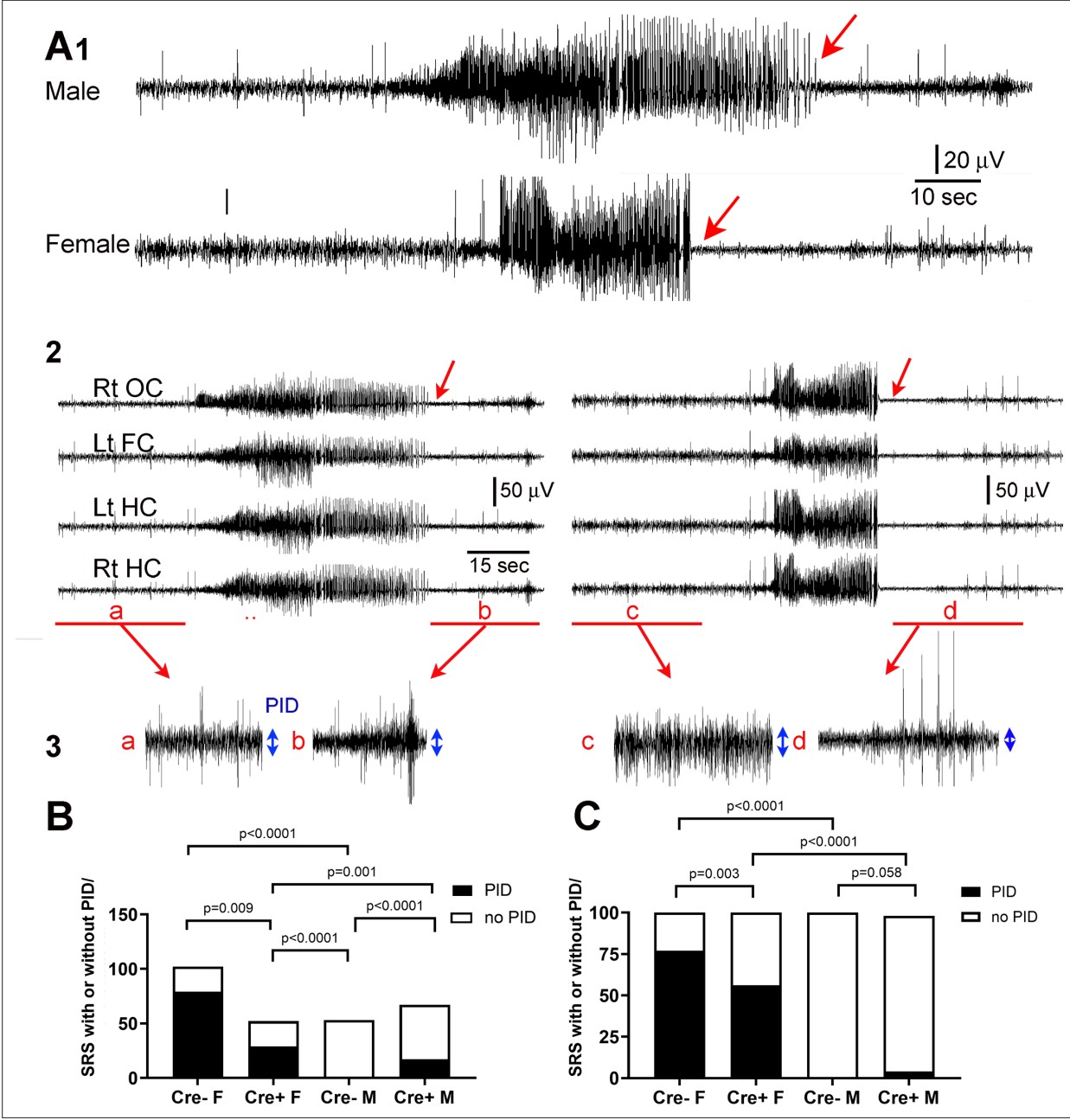

**Figure 3.** Reduced postictal depression in Cre+ female mice. (**A**) (1) A seizure of a male mouse and a female mouse are shown to illustrate the end of the seizure (red arrow). (2) All four channels are shown for the male (left) and female mouse (right). Rt OC, right occipital cortex; Lt FC, left frontal cortex; Lt HC, left hippocampus; Rt HC, right hippocampus. The seizure is shown on a compressed scale. (3) The areas in A2 marked by the red bar are expanded. The blue double-sided arrows reflect the mean EEG amplitude before (**a, c**) and after the seizure (**b, d**). The female seizure shows postictal depression (PID). (**B**) For all spontaneous recurrent seizures (SRS) in the 3 week-long recording period (n=5 mice/group), there were significant differences between groups, with the number of SRS with PID greater in females (Cre- females vs. Cre- males). PID was reduced in Cre+ females compared to Cre- females. All comparisons were Fisher's exact tests. (**C**). The same data are plotted but the percentages are shown instead of the numbers of seizures. Fisher's exact tests showed similar group differences except for Cre+ and Cre- males which only showed a trend. Together the data show females exhibited more PID than males and Cre+ mice were protected, especially females.

(108/154 seizures, 70.5%) than all males (17/120 seizures, 14.2%; Fisher's exact tests, p≤0.0001, *Figure 3B-C*). Postictal depression in males was relatively rare and genotypic differences were not robust (*Figure 3B-C*).

### Clusters of seizures

Next, we asked if the distribution of seizures during the 3 weeks of video-EEG was affected by increasing adult-born neurons. In *Figure 4A, a* plot of the day-to-day variation in seizures is shown with each day of recording either black (if there were seizures) or white (if there were no seizures).

The number of days with seizures were similar between genotypes (Cre-: 8.6±0.6 days, n=18; Cre+: 8.4±0.6 days, n=17; Student's t-test, t(33)=0.23, p=0.822; *Figure 4B1*). The number of consecutive days without seizures, called the seizure-free interval, was also similar between genotypes (Cre-: 6.4±0.4 days, n=19; Cre+: 7.7±0.6 days, n=17; Student's t-test, t(34)=1.65, p=0.107; *Figure 4B2*). When data were segregated based on sex, a two-way ANOVA showed no effect of genotype (F(1,32)=1.18, p=0.286) or sex on the number of days with seizures (F(1,32)=0.86, p=0.361; *Figure 4B3*). There also was no effect of genotype (F(1,32)=2.86, p=0.100) or sex (F(1,32)=0.53, p=0.471) on seizure-free interval (*Figure 4B4*).

Clustering is commonly manifested in consecutive days with frequent seizures. Clusters of seizures can have a substantial impact on the quality of life (*Haut, 2015*; *Jafarpour et al., 2019*) so they are important. In humans, clusters are defined as at least 3 seizures within 24 hr (*Goffin et al., 2007*; *Jafarpour et al., 2019*). Therefore, we defined clusters as >1 consecutive day with ≥3 seizures/day (*Figure 4C*). The duration of clusters were similar between genotypes (Cre-: 3.8±0.7 days, n=19; Cre+: 3.0±0.3 days, n=18; Mann-Whitney's *U* test, *U* statistic 159, p=0.723; *Figure 4D1*). Next, we calculated the number of days between clusters, which we call the intercluster interval. Genotypes were similar (Cre-: 7.2±0.8 days, n=13; Cre+: 9.2±0.8 days, n=9; Student's t-test, t(20)=1.70, p=0.104; *Figure 4D2*).

Two-way ANOVA was then performed on data segregated by sex. For cluster duration, there was no effect of genotype (F(1,33)=3.36, p=0.076) but there was a main effect of sex (F(1,33)=7.66, p=0.009) and a significant interaction of genotype and sex (F(1,33)=0.66, p=0.009). Cre+ females had fewer days with ≥3 seizures than Cre- females (Cre- females: 6.3±1.4 days; Cre+ females: 3.0±0.4 days; Tukey's post-hoc test, p=0.009; *Figure 4D3*). These data suggest Cre+ females were protected from the peak of a cluster, when seizures increase above 3 /day.

There was no effect of genotype (F(1,17)=2.72, p=0.117) or sex (F(1,17)=2.72, p=0.117) on the intercluster interval (*Figure 4D4*). However, this result may have underestimated effects because Cre+ females often had such a long interval that it was not captured in the 3-week-long recording period. That led to fewer Cre+ females that were included in the measurement of intercluster interval. In 5 out of 9 (i.e. 55%) Cre+ females, there was only one cluster in 3 weeks, so intercluster interval was too long to capture. Of those mice where intercluster interval could be measured, Cre- females had an interval of 5.7±1.0 days (n=7) and Cre+ females had a 9.0±1.1 day interval (n=4). That difference was not significant.

In summary, Cre+ females had fewer seizures, fewer days with ≥3 seizures, reduced postictal depression, and appeared to have a long period between clusters of seizures.

## Before and after epileptogenesis, Cre+ female mice exhibited more immature neurons than Cre- female mice but that was not true for male mice

### Prior to SE

We first confirmed that prior to pilocarpine treatment, Cre+ mice had more young adult-born neurons compared to Cre- mice (*Figure 5A*, *Figure 5—figure supplement 1A–D*). To that end, we quantified the adult-born GCs associated with the GCL/SGZ in both Cre+ and Cre- mice. DCX was used as a marker because it is highly expressed in immature neurons (*Brown et al., 2003*; *Couillard-Despres et al., 2005*). The area of the GCL/SGZ that exhibited DCX-ir was calculated and expressed as a percent of the total area of the GCL/SGZ (*Figure 5C*).

A two-way ANOVA with sex and genotype as factors showed a significant effect of genotype (F(1,9)=60.78, p<0.001) but not sex (F(1,9)=1.20, p=0.301; *Figure 5D*). Post-hoc comparisons showed that Cre+ females had more DCX than Cre- females (p=0.001) and the same was true for males

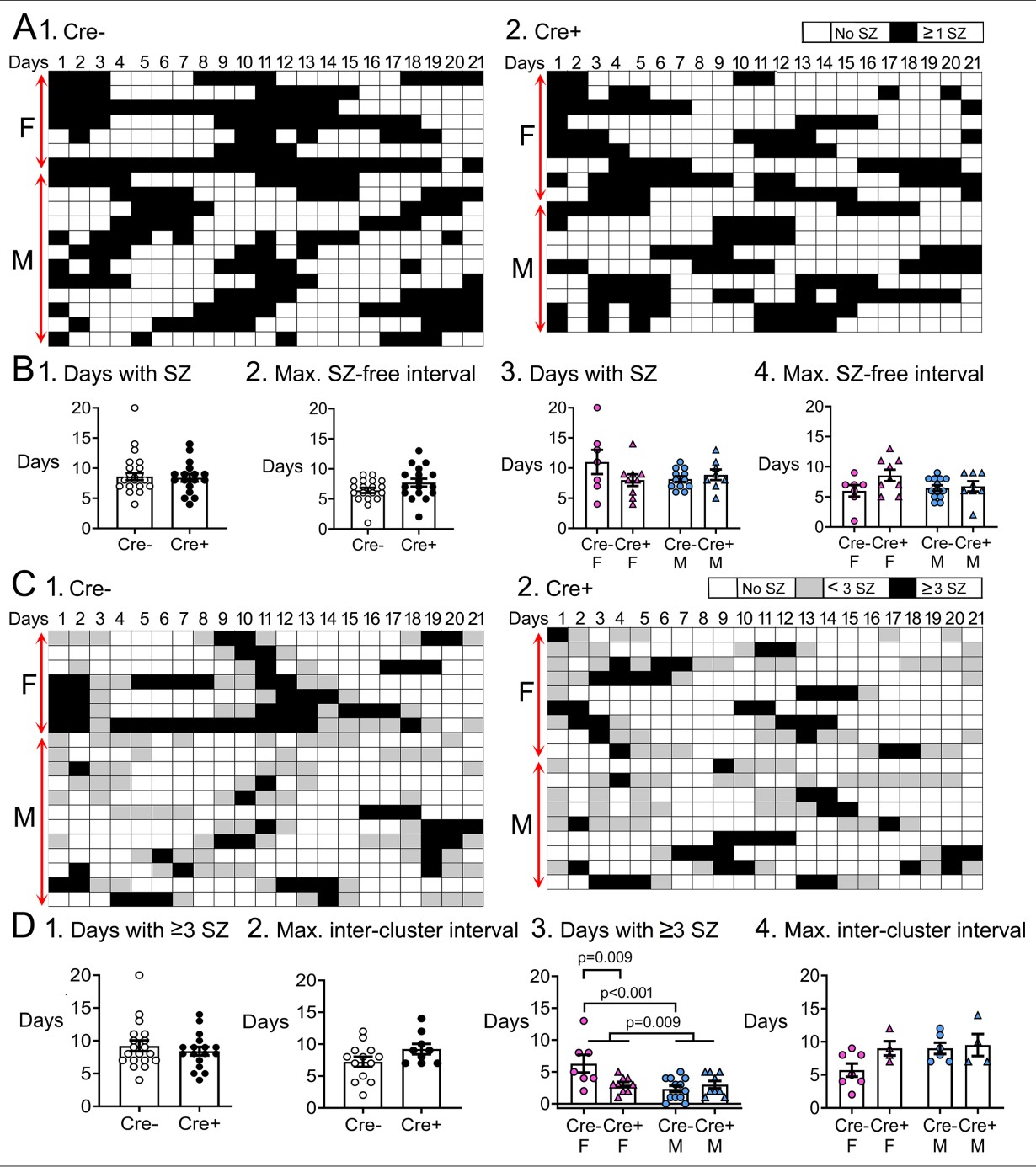

**Figure 4.** Temporal dynamics of chronic seizures. (**A**) Each day of the 3-weeks-long EEG recording periods are shown. Each row is a different mouse. Days with seizures are coded as black boxes and days without seizures are white. (**B**) (1) The number of days with seizures were similar between genotypes (t-test, p=0.822). (2) The maximum seizure-free interval was similar between genotypes (t-test, p=0.107). After separating females and males, two-way ANOVA showed no effect of genotype or sex on days with seizures. (4) Two-way ANOVA showed no effect of genotype or sex on the maximum seizure-free interval. (**C**) Then same data are shown but days with >3 seizures are black, days with <3 seizures as gray, and are white. Clusters of seizures are reflected by the consecutive black boxes. (**D**) The cluster durations were similar between genotypes (Mann-Whitney's *U* test, p=0.723). (1) The maximum inter-cluster interval was similar between genotypes (t-test, p=0.104). (2) Cre+ females had significantly fewer clusters than Cre- females (two-way ANOVA followed by Tukey's post-hoc test, p=0.009). There was a sex difference, with females having more clusters than males (p=0.009). Cre- females had more days with >3 seizures than control males (Cre- females: 6.3±1.4 days; Cre- males: 2.3±0.5 days; Tukey's post-hoc test, p<0.001). (3) There was no significant effect of genotype or sex on the maximum inter-cluster interval. However, there was a trend for the inter-cluster interval to be longer in Cre+ females relative to Cre- females.

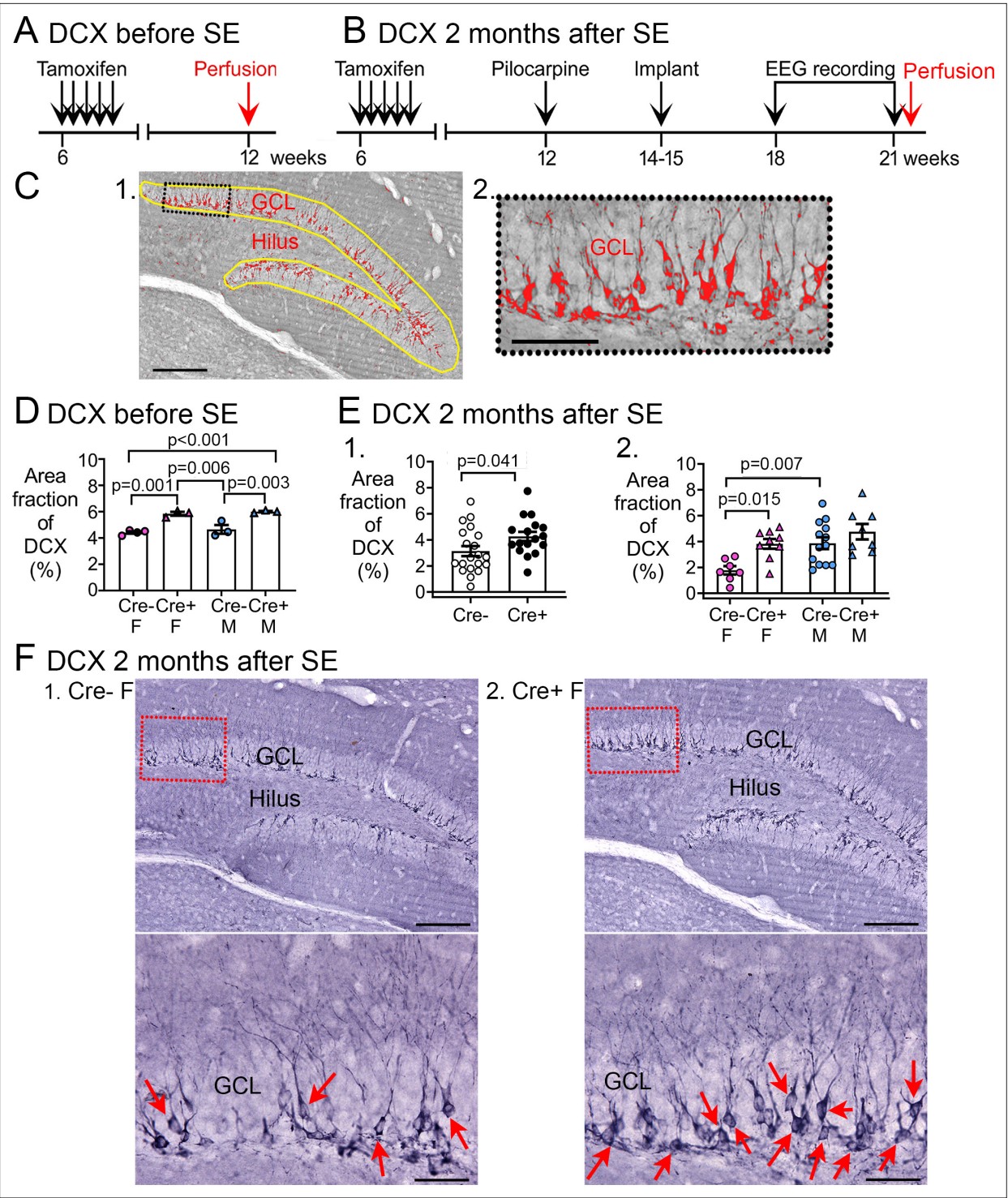

**Figure 5.** Increased Doublecortin (DCX) in Cre+ mice. (**A–B**) The experimental timelines are shown. (**A**) Mice were perfusion-fixed 6 weeks after tamoxifen injection, just before SE. Sections were then stained for DCX. (**B**) Mice were tested 2 months after SE, after EEG recording. Then mice were perfused and staining was conducted for DCX. (**C**) (1) DCX quantification. DCX-ir within a region of interest (ROI; yellow lines) including the subgranular zone (SGZ) and GCL was thresholded. DCX-ir above the threshold is shown in red. Calibration, 100 μm. (2) The inset is expanded to the right. Calibration, 50 μm. (**D**) The area of DCX-ir relative to the area of the ROI (referred to as area fraction) was greater in Cre+ mice compared to Cre- mice. Two-way ANOVA followed by Tukey's post-hoc tests, all p<0.05. (**E**) Cre+ mice had increased DCX-ir relative to Cre- mice 2 months after SE. (1) Sexes were pooled. The area fraction of DCX-ir was greater in Cre+ than Cre- mice (t-test, p=0.041). (2) When sexes were separated, Cre+ females showed greater DCX-ir than Cre- females (two-way ANOVA followed by Tukey's post-hoc test, p=0.015). There was a sex difference, with Cre- males showing

*Figure 5 continued on next page*

*Figure 5 continued*

more DCX-ir than Cre- females (p=0.007). DCX-ir was similar in Cre- and Cre+ males (p=0.498). (**F**) Representative examples of DCX-ir 2 months after SE. (1) Cre- female mouse. (2) Cre+ female mouse. The red boxes in a are expanded in b. Arrows point to DCX-ir cells. Calibration, 100 μm (**a**); 50 μm (**b**).

The online version of this article includes the following figure supplement(s) for figure 5:

**Figure supplement 1.** Doublecortin (DCX) in Cre- and Cre+ mice before SE.

(p=0.003). Cre+ males had more DCX than Cre- females (p<0.001), and Cre+ females had more DCX than Cre- males (p=0.006). Cre- females and males were not different (p=0.774). The results are consistent with studies using the same methods which showed that Cre+ males have more DCX compared to Cre- males (*Jain et al., 2019*). Together the data suggest that Cre+ mice had more young adult-born neurons than Cre- mice immediately before SE.

## After epileptogenesis

We also quantified DCX at the time when epilepsy had developed, after the 3-week-long EEG recording (*Figure 5B*). Representative examples of DCX expression in the GCL/SGZ are presented in *Figure 5F* and *Figure 5E* shows the area fraction of DCX in the GCL/SGZ was significantly greater in Cre+ mice than Cre- mice (Cre-: 3.1±0.4%, n=20; Cre+: 4.2±0.3%, n=17; Student's t-test, t(35)=2.13, p=0.041; *Figure 5D1*). Therefore, Cre+ mice had increased DCX in the GCL/SGZ after chronic seizures had developed.

To investigate a sex difference, a two-way ANOVA was conducted with genotype and sex as main factors. There was a significant effect of genotype (F(1,33)=12.62, p=0.001) and sex (F(1,33)=11.68, p=0.002), with Cre+ females having more DCX than Cre- females (Cre- female: 1.8±0.3, n=7; Cre+ female: 3.8±0.4, n=9; Tukey's post-hoc test, p=0.015; *Figure 5E*). In contrast, DCX levels were similar between Cre+ and Cre- male mice (p=0.498, *Figure 5E*). Therefore, elevated DCX occurred after chronic seizures had developed in Cre+ mice but the effect was limited to females. Because Cre+ epileptic females had increased immature neurons relative to Cre- females at the time of SE, and prior studies show that Cre+ females had less neuronal damage after SE (*Jain et al., 2019*), female Cre+ mice might have had reduced chronic seizures because of high numbers of immature neurons. However, the data do not prove a causal role.

It is notable that the Cre+ male mice did not show increased numbers of immature neurons at the time of chronic seizures but Cre+ females did. It is possible that there was a 'ceiling' effect in DCX expression that would explain why male Cre+ mice did not have a significant increase in immature neurons relative to male Cre- mice.

## Hilar ectopic granule cells

Based on the literature showing that reducing hilar ectopic GCs decreases chronic seizures after pilocarpine-induced SE (*Cho et al., 2015*), we hypothesized that female Cre+ mice would have fewer hilar ectopic GCs than female Cre- mice. However, female Cre+ mice did not have fewer hilar ectopic GCs.

To quantify hilar ectopic GCs we used Prox1 as a marker. Prox1 is a common marker of GCs in the GCL (*Pleasure et al., 2000*; *Galeeva et al., 2007*; *Galichet et al., 2008*; *Steiner et al., 2008*; *Iwano et al., 2012*), and the hilus (*Scharfman et al., 2007a*; *Hester and Danzer, 2013*; *Cho et al., 2015*; *Bermudez-Hernandez et al., 2017*).

Cre+ mice had significantly more hilar Prox1 cells than Cre- mice (Cre-: 19.6±1.9 cells, n=18; Cre+: 60.5±7.9 cells, n=18; Student's t-test, t(34)=5.76, p<0.001; *Figure 6C1*). A two-way ANOVA with genotype and sex as main factors showed no effect of sex (F(1,32)=0.28, p=0.595) but a significant effect of genotype (F(1,32)=0.23, p<0.0001) with more hilar Prox1 cells in female Cre+ than female Cre- mice (Cre- female: 18.2±3.3 cells, n=7; Cre+ female: 57.0±8.3 cells, n=9; Tukey's post-hoc test, p<0.001; *Figure 6C2*) and the same for males (Cre- male: 20.4±2.4 cells, n=11; Cre+ male: 63.9±14.0 cells, n=9; p=0.001; *Figure 6C2*).

In past studies, hilar ectopic GCs have been suggested to promote seizures (*Scharfman et al., 2000*; *Jung et al., 2006*; *Cho et al., 2015*). Therefore, we asked if the number of hilar ectopic GCs correlated with the number of chronic seizures. When Cre- and Cre+ mice were compared (both sexes pooled), there was a correlation with number of chronic seizures (*Figure 6D1*) but it suggested that

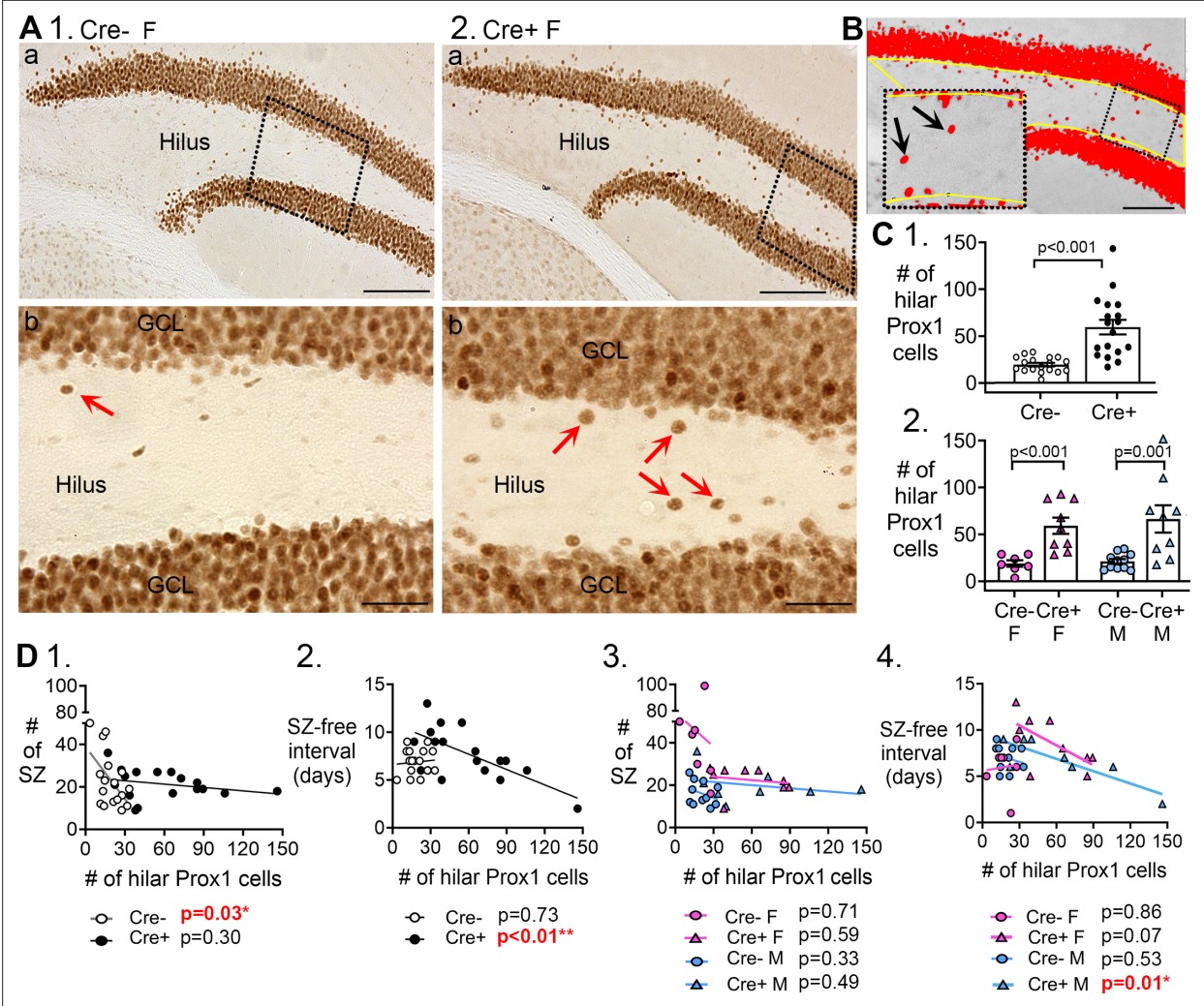

**Figure 6.** Hilar Prox1-ir cells increased in Cre+ mice. (**A**) Representative examples of hilar Prox1-ir in Cre- (1) and Cre+ (2) mice are shown. The boxes in a are expanded in b. Arrows point to hilar Prox1-ir cells, corresponding to hilar ectopic GCs. Calibration, 100 μm (a); 50 μm (b). (**B**) Prox1-ir is shown, within a hilar region of interest (ROI). The area of the ROI above the threshold, relative to the area of the ROI, is red. This area is called the area fraction, and was used to quantify hilar Prox1-ir. Calibration, 100 μm. (**C**) (1) Cre+ mice had more hilar Prox1-ir cells than Cre- mice (t-test, p<0.001). (2) When sexes were divided, Cre+ mice had more hilar Prox1-ir cells than Cre- mice in both female (two-way ANOVA followed by Tukey's post-hoc test, p<0.001) and male mice (p=0.001). (**D**) Correlations between hilar Prox1-ir cells and measurements of chronic seizures. (1) All Cre- and Cre- mice were compared regardless of sex. For the Cre- mice there was a significant inverse correlation between the # of Prox1-ir cells and # of chronic seizures (R²=0.296). Thus, the more Prox1-ir cells there were, the fewer chronic seizures there were. However, that was not true for Cre+ mice (R²=0.072). (2) There was an inverse correlation between the number of hilar Prox1-ir cells and the seizure-free interval for Cre+ mice (R²=0.467) but not Cre- mice (R²=0.008). Thus, the more hilar Prox1-ir cells there were, the shorter the seizure-free periods were. However, this was not true for Cre- mice. (3) When data were divided by genotype and sex there was no significant correlation between hilar Prox1-ir cells and # of seizures (Cre- F, R²=0.0035; Cre+ F, R²=0.043; Cre- M, R²=0.104; Cre+ M, R²=0.083). (4) When data were divided by genotype and sex, there was a significant inverse correlation for the # of hilar Prox1-ir cells and seizure-free interval, but only for male Cre+ mice (R²=0.704). Cre+ females showed a trend (R²=0.395) and Cre- mice did not (Cre- F, R²=0.007, Cre- M, R²=0.046).

more hilar ectopic GCs improved rather than worsened seizures. However, the correlation was only in Cre- mice, and when sexes were separated there was no correlation (*Figure 6D3*).

When seizure-free interval was examined with sexes pooled, there was a correlation for Cre+ mice (*Figure 6D2*) but not Cre- mice. Strangely, the correlations of Cre+ mice with seizure-free interval (*Figure 6D2 and D4*) suggest ectopic GCs shorten the seizure-free interval and, therefore, worsen epilepsy, opposite of the correlative data for numbers of chronic seizures. In light of these inconsistent results, it seems that hilar ectopic granule cells had no consistent effect on chronic seizures.

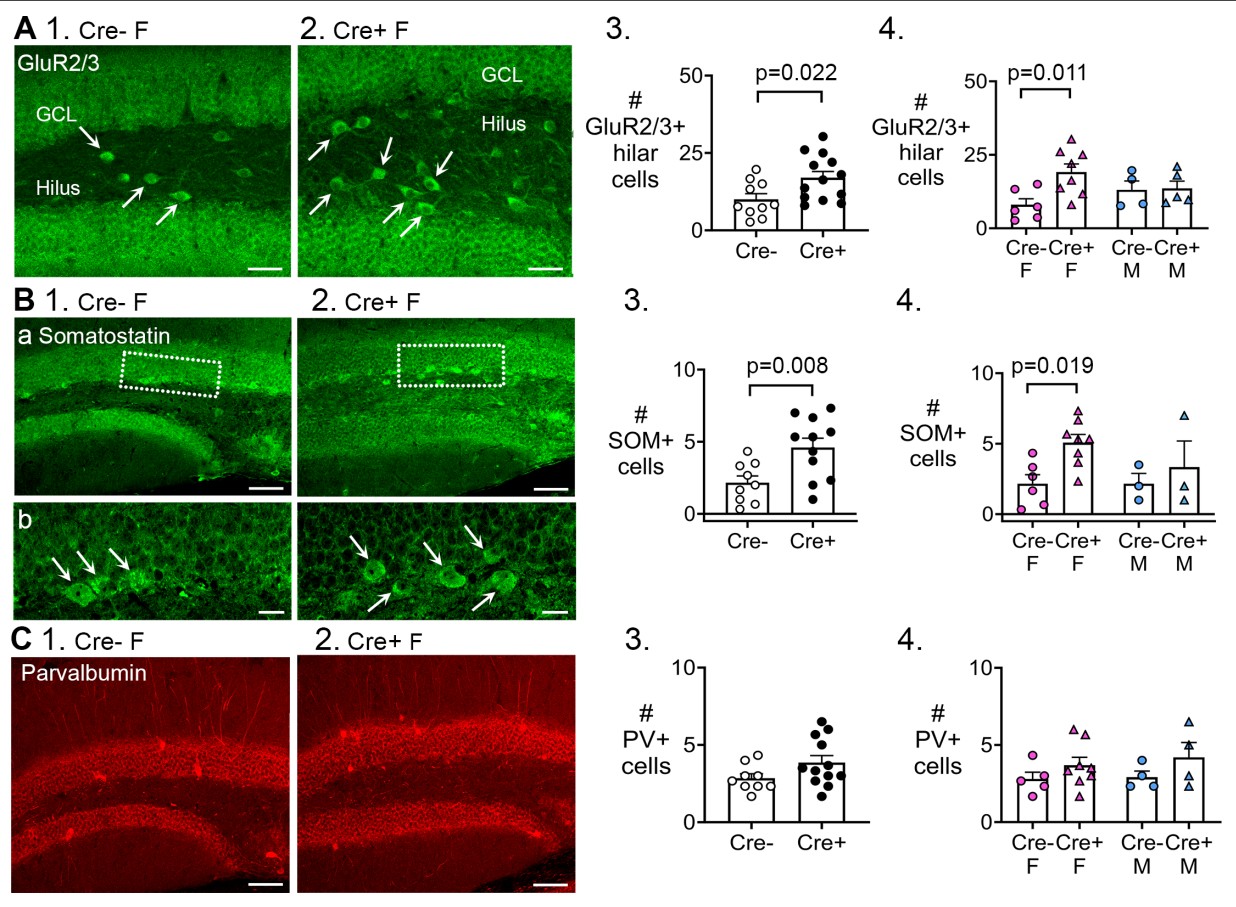

**Figure 7.** Preserved mossy cells and hilar somatostatin (SOM) cells in Cre+ female mice but not parvalbumin interneurons. (**A, 1–2**) Representative examples of GluR2/3 labeling of Cre- (1) and Cre+ mice (2). Calibration, 50 µm. (3) Cre+ mice had more hilar GluR2/3-immunofluorescent (positive; +) cells than Cre- mice (t-test, p=0.022). Sexes were pooled. (4) After separating females and males, Cre+ females showed more hilar. GluR2/3 + cells than Cre- females (two-way ANOVA followed by Tukey's post-hoc test, p=0.011). Hilar GluR2/3+ cells were similar between genotypes in males (p=0.915). (**B, 1–2**) Representative examples of SOM labeing in Cre- and Cre+ mice are shown. Calibration, 100 µm (**a**); 20 µm (**b**). (3) In pooled data, Cre+ mice had more hilar SOM cells than Cre- mice (t-test, p=0.008). (4) After separating females and males, Cre+ females showed more hilar SOM cells than Cre- females (p=0.019). Hilar SOM cells were similar between genotypes in males (p=0.897). (**C, 1–2**) Representative examples of parvalbumin labeling in Cre- and Cre+ mice are shown. Calibration, 100 µm. (3) The number of parvalbumin+ cells in the dentate gyrus (DG) were similar in Cre- and Cre+ mice in pooled data (t-test, p=0.095). (4) There was no effect of genotype (p=0.096) or sex (p=0.616) on the number of DG parvalbumin+ cells.

The online version of this article includes the following figure supplement(s) for figure 7:

**Figure supplement 1.** Additional analyses of GluR2/3, somatostatin (SOM), and parvalbumin-expressing cells.

## Increased adult-born neurons preserve mossy cells and hilar SOM interneurons but have little effect on parvalbumin interneurons

It has been suggested that epileptogenesis after a brain insult like SE is due to the hippocampal damage caused by the insult (*Cavalheiro et al., 1996*; *Herman, 2002*; *Mathern et al., 2008*; *Dudek and Staley, 2012*; *Dingledine et al., 2014*). Therefore, one of the reasons why increasing adult-born neurons reduced chronic seizures could be that it reduced neuronal damage after SE. Indeed, increasing adult-born neurons reduces neuronal damage after SE (*Jain et al., 2019*). Here, we examined the loss of vulnerable hilar mossy cells and SOM cells because they have been suggested to be critical (*Sloviter, 1987*; *Cavazos and Sutula, 1990*; *Cavazos et al., 1994*; *Henshall and Meldrum, 2012*; *Huusko et al., 2015*). We asked whether Cre+ mice had preserved mossy cells (*Figure 7A*) and SOM neurons (*Figure 7B*). For comparison, we quantified the relatively seizure-resistant parvalbumin-expressing GABAergic neurons (*Figure 7C*). An antibody to GluR2/3 was used as a marker of mossy cells (*Leranth et al., 1996*) and a SOM antibody for SOM cells (*Leranth et al., 1990*; *Savanthrapadian et al., 2014*; *Botterill et al., 2019*).

The results showed that Cre+ mice had more GluR2/3-expressing hilar cells than Cre- mice (Cre-: 10.0±1.8 cells, n=10; Cre+: 17.0±2.0 cells, n=13; Student's t-test, t(21)=2.46, p=0.022; Fig. 7A1-3). We confirmed that the GluR2/3+ hilar cells were not double-labeled with Prox1, suggesting they corresponded to mossy cells, not hilar ectopic GCs (*Figure 7—figure supplement 1A*). To investigate sex differences, a two-way ANOVA was conducted with genotype and sex as the main factors. There was a significant effect of genotype (F(1,18)=4.95, p=0.039) with Cre+ females having more GluR2/3 cells than Cre- females (Cre- female: 8.0±2.0 cells, n=6; Cre+ female: 19.1±2.7 cells, n=8; Tukey's post-hoc test, p=0.011; *Figure 7A4*). GluR2/3-ir hilar cells were similar in males (Cre- male: 13.0±3.0 cells, n=4; Cre+ male: 13.6±2.5 cells, n=5; p=0.915; *Figure 7A4*). These results in dorsal DG also were obtained in ventral DG (*Figure 7—figure supplement 1B-C*). The data suggest that having more GluR2/3-ir mossy cells could be a mechanism that allowed Cre+ females to have reduced chronic seizures compared to Cre- females. Equal numbers of GluR2/3 mossy cells in Cre+ and Cre- males could relate to their lack of protection against chronic seizures.

Next, we measured SOM hilar cells in pooled data (females and males together). These results were analogous to the data for GluR2/3, showing that Cre+ mice had more hilar SOM cells than Cre- mice (Cre-: 2.1±0.5 cells, n=9; Cre+: 4.6±0.6 cells, n=11; Student's t-test, t(18)=2.95, p=0.008; Fig. 7B1-3). When the sexes were separated, a two-way ANOVA showed a significant effect of genotype (F(1,16)=5.14, p=0.038) and no effect of sex (F(1,18)=0.94, p=0.346). However, Cre+ females had more SOM cells than Cre- females (Cre- female: 2.2±0.6 cells, n=6; Cre+ female: 5.1±0.6 cells, n=8; Tukey's post-hoc test, p=0.019; *Figure 7B4*), although only in dorsal DG (*Figure 7B4*) not ventral DG (*Figure 7—figure supplement 1C*). Numbers of SOM cells were similar in males (Cre- male: 2.2±0.7 cells, n=3; Cre+ male: 3.3±1.8 cells, n=3; p=0.897; *Figure 7B4*) in both dorsal and ventral DG (*Figure 7—figure supplement 1B-C*). Therefore, the ability to preserve more mossy cells and SOM hilar cells in Cre+ females could be a mechanism by which Cre+ females were protected from chronic seizures.

Parvalbumin-ir cells were not significantly different between genotypes (Student's test, t(19)=1.76, p=0.095; Fig. 7C1-3). A two-way ANOVA showed no effect of genotype (F(1,17)=3.10, p=0.096) or sex (F(1,17)=0.26, p=0.616) on the numbers of parvalbumin cells. The results were the same in dorsal and ventral DG (*Figure 7—figure supplement 1B-C*). These data are consistent with the idea that loss of parvalbumin-expressing cells has not been considered to play a substantial in epileptogenesis in the past (*Sloviter, 1987*; *Sloviter, 1994*). However, it should be noted that subsequent research has shown that the topic is complicated because parvalbumin expression may decline even if the cells do not die (*André et al., 2001*; *Sun et al., 2007*) and data vary depending on the animal model (*van Vliet et al., 2004*; *Huusko et al., 2015*).

## VI. Increased adult-born neurons decreased neuronal damage after SE

In our previous study of Cre+ and Cre- mice (*Jain et al., 2019*), tamoxifen was administered at 6 weeks and SE was induced at 12 weeks (like the current study). We examined neuronal loss 3 days after SE, when neuronal loss in the hilus and area CA3 is robust in wild-type mice. There is also some neuronal loss in CA1 at 3 days but more at 10 days after SE. We found less neuronal loss in Cre+ mice in these three areas (*Jain et al., 2019*). In the current study, we examined 10 days after SE (*Figure 8A–B*) because at this time delayed neuronal loss occurs, providing a better understanding of CA1 and the subiculum because delayed cell death occurs there. The intent was to determine if Cre+ mice exhibited less neuronal loss or not in CA1 and the subiculum.

To quantify Fluorojade C, ROIs were drawn digitally around the pyramidal cell layers (*Figure 8B*). As shown in *Figure 8C*, there was less Fluorojade C staining in Cre+ female mice relative to Cre- female mice in both CA1 and the subiculum. The area of the ROI that showed Fluorojade C-positive cells was calculated as area fraction and expressed as % in *Figure 8D*. For females, a two-way ANOVA with genotype and subfield as factors showed a significant effect of genotype (F(1,14)=11.21, p=0.005) with a smaller area fraction in Cre+ mice than Cre- mice for subiculum (p=0.032) and a trend in CA1 (p=0.060; *Figure 8D1*). Males showed no significant differences either in genotype (F(1,8)=0.02, p=0.872) or subfield (F(1,8)=0.16, p=0.698; *Figure 8D2*). When genotypes were pooled, females were not different than males (two-way ANOVA, sex, F(1,32)=2.28, p=0.140) in either subfield (F(1,32)<0.001; p=0.997; *Figure 8D3*). Thus, Cre+ female mice were protected from damage relative to Cre- females.

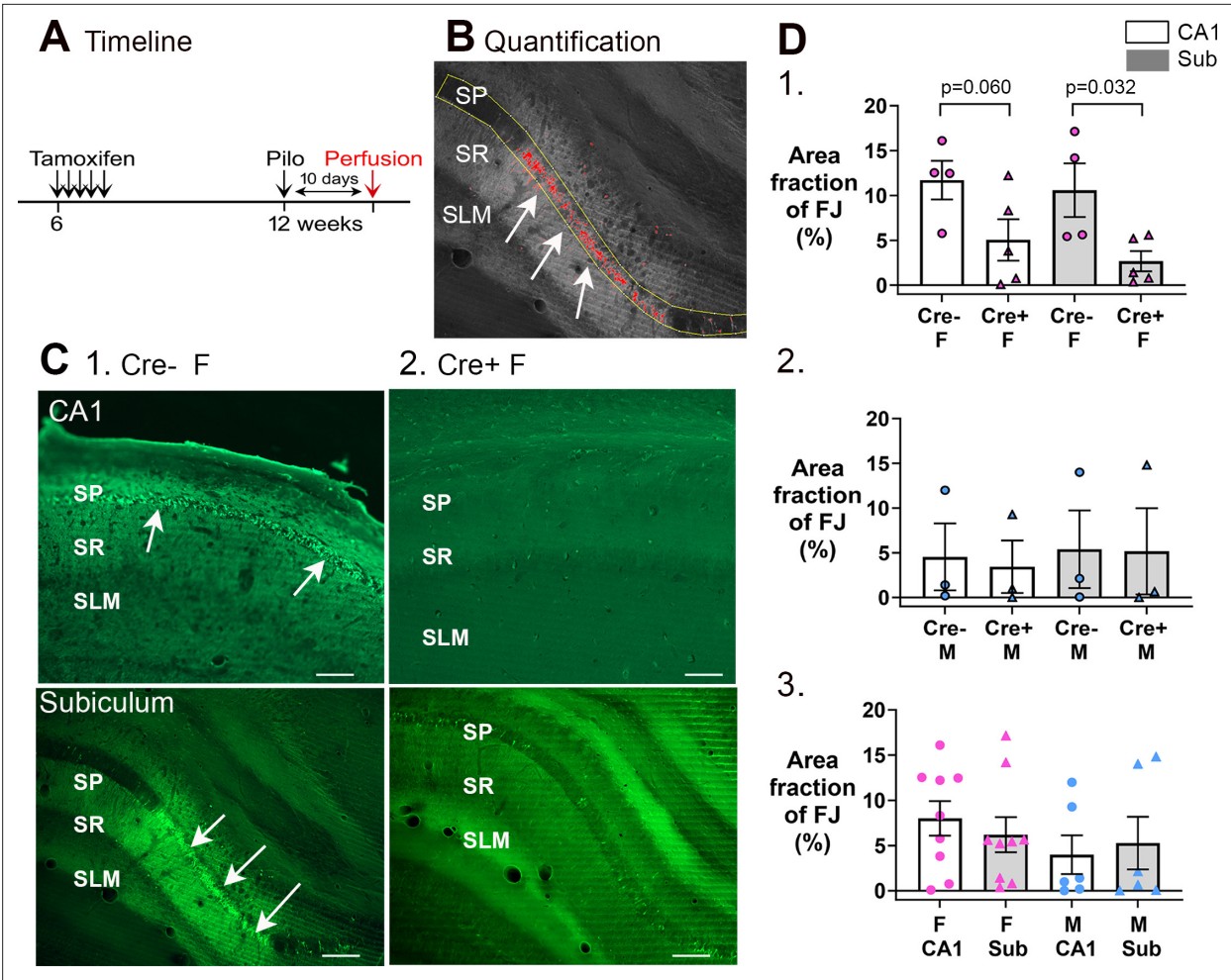

**Figure 8.** Cre+ female mice had less neuronal loss in hippocampus after SE. (**A**) A timeline is shown to illustrate when mice were perfused to examine Fluorojade C staining. All mice were perfused 10 days after SE, a time when delayed cell death occurs after SE, mainly in area CA1 and subiculum. Note that prior studies showed hilar and CA3 neurons, which exhibit more rapid cell death after SE, are protected from cell loss in Cre+ mice examined 3 days after SE (*Jain et al., 2019*). Also, there was protection of CA1 for 3 days (*Jain et al., 2019*). (**B**) Quantification. Fluorojade C was thresholded using ImageJ and the pyramidal cell layer outlined in yellow. The fraction above threshold relative to the entire region of interest (ROI) (area fraction) was calculated (see Methods). (**C**) Examples of Fluorojade C staining in CA1 (top) and subiculum (bottom) of Cre+ female (1) and Cre- female (2) mice. SO, stratum oriens; SP, stratum pyramidale; SR, stratum radiatum; SLM, stratum lacunosum-moleculare. Arrows point to numerous Fluorojade C-stained neurons in Cre- mice but not Cre+ mice. Calibration, 200 μm. (**D** ) (1) Comparisons of female mice by two-way ANOVA showed less Fluorojade C in Cre+ mice for subiculum (p=0.032). (2) Comparisons of male mice showed no significant effect of genotype on Fluorojade C ($F_{(1,8)}$=0.16, p=0.698) in either subfield ($F_{(1,8)}$=0.02, p=0.872). (3) When genotypes were pooled, female mice did not have significantly more damage than males (two-way ANOVA, $F_{(1,32)}$=2.28, p=0.149) and there was no effect of subfield ($F_{(1,32)}$<0.001, p=0.997).

## Discussion

This study showed that conditional deletion of *Bax* from Nestin-expressing progenitors increased young adult-born neurons in the DG when studied 6 weeks after deletion and using DCX as a marker of immature neurons. In a different set of mice, pilocarpine was used to induce epileptogenesis. The chronic seizures, measured 4–7 weeks after pilocarpine, were reduced in frequency by about 50% in females. Therefore, increasing young adult-born neurons before the epileptogenic insult can protect against epilepsy. However, we do not know if the protective effect was due to the greater number of new neurons before SE or other effects. Past data would suggest that increased numbers of newborn neurons before SE lead to a reduced SE duration and less neuronal damage in the days after SE (*Jain et al., 2019*). That would be likely to lessen the epilepsy after SE. However, there may have been additional effects of larger numbers of newborn neurons prior to SE.

Increasing young adult-born neurons has been shown to protect the hippocampus from SE-induced neuronal loss (*Jain et al., 2019*) which is a major contributor to epileptogenesis. Therefore, by protecting against SE-induced neuronal loss the young adult-born neurons could have reduced the severity of epilepsy. Indeed, we showed that the Cre+ female mice that had reduced chronic seizures had preservation of hilar mossy cells and SOM cells, two populations that are lost in SE-induced epilepsy and considered to contribute to epileptogenesis.

There were major surprises in the current study. First, the results are unexpected because suppressing adult-born neurons was shown to reduce chronic seizures (*Cho et al., 2015*). Here, increasing adult-born neurons did not have the opposite effect.

It was also unanticipated that only females with *Bax* deletion showed a significant increase in young adult-born neurons and a significant reduction in chronic seizures. The larger effect of increasing adult-born neurons in female mice may be attributable to sex differences in Bax (discussed below). The other remarkable finding relates to hilar ectopic GCs. These GCs have been suggested to promote epilepsy (*Scharfman, 2004*; *Jung et al., 2006*; *Scharfman et al., 2007a*; *Parent and Murphy, 2008*; *Hester and Danzer, 2013*; *Cho et al., 2015*), but hilar ectopic GCs increased in females with reduced seizures. The association of more hilar ectopic GCs with fewer chronic seizures was unexpected.

## Effects of *Bax* deletion on SE

In past studies, suppressing adult-born neurons made kainic acid-induced SE worse, and pilocarpine-induced SE was also worse (*Iyengar et al., 2015*; *Jain et al., 2019*). In the present study, SE was affected also. The duration of SE was reduced in Cre+ mice. In the Cre+ females, however, it was unclear. Nevertheless, the first seizure after pilocarpine injection was often less severe in Cre+ females, and power showed a tendency to be reduced during SE. Therefore, SE might have been less severe in the Cre+ females, and this could have contributed to reduced neuronal loss and chronic seizures.

## Chronic seizures

It is remarkable that increasing adult-born neurons for 6 weeks was sufficient to reduce seizures long-term. It is consistent with the idea that normally the young adult-born neurons inhibit other GCs, which supports the DG gate function (*Hsu, 2007*; *Drew et al., 2016*). This gate has been suggested to be an inhibitory barrier to entry of seizures from cortex into hippocampus (*Coulter and Carlson, 2007*; *Hsu, 2007*; *Krook-Magnuson et al., 2015*). That entry is deleterious because seizures that pass from entorhinal cortex to the GCs and then CA3 are likely to continue to CA1 and back to cortex, causing reverberatory (long-lasting, severe) seizures. The reason for the relatively ease of reverberation once past the DG gate is that the synapses between GCs and CA3, CA1, and cortex are excitatory. Also, the GCs have especially powerful excitatory synapses on CA3 pyramidal cells (*Henze et al., 2000*; *Scharfman and MacLusky, 2014*), although these are normally mitigated by GABAergic circuitry (*Acsády et al., 1998*).

These data are consistent with the demonstration that adult-born neurons protect against other pathological conditions such as Alzheimer's disease (*Choi et al., 2018*; *Choi and Tanzi, 2019*). However, it is important to note that all effects are unlikely to be mediated only by the DG. The olfactory bulb and other areas also have adult-born neurons and they could contribute to epilepsy, especially those epilepsy syndromes with mechanisms that are extrahippocampal.

## Clusters of seizures

There were fewer days with >3 seizures in Cre+ female mice which is another way that Cre+ females were protected from chronic seizures. These findings are valuable because clusters in humans have a significantly negative impact on health and quality of life (*Haut, 2015*; *Jafarpour et al., 2019*).

The results may have underestimated the effect on clusters because we did not measure the interval between clusters in many Cre+ female mice. The reason is that the interval between clusters increased in some mice so they only had one cluster in 3 weeks. Thus intercluster interval appeared to lengthen in Cre+ females but animals with only one cluster had to be excluded. In the end, the results were not statistically significant.

## Sex differences

Females showed more of an effect of conditional *Bax* deletion than males. Insight into this sex difference came when the same assessments were made before SE because at that time there was no sex difference. Cre+ females had more adult-born neurons than Cre- females and Cre+ males had more than Cre- males. In addition, the levels of DCX were similar in Cre+ females and Cre+ males.

However, after epilepsy developed, there was a sex difference. Cre- females had less DCX than Cre-males. One explanation is that cell birth during epileptogenesis was greater in males because it is in the developing hippocampus (*Sisk et al., 2016*) and SE has been suggested to rekindle developmental programs (*Ben-Ari and Holmes, 2006*). Another possibility is males had less programmed cell death during epileptogenesis.

Indeed during development, females have more apoptotic profiles than males and the sex difference was blocked by *Bax* deletion (*Forger et al., 2004*). A final possibility is that cell death during epileptogenesis is *Bax*-dependent in females but *Bax*-independent in males. Support for this idea comes from studies of ischemic cell death, which is caspase-dependent in females but not males (*Siegel and McCullough, 2011*).

## Hilar ectopic GCs

In the normal brain, adult-born neurons in the DG are thought to arise mainly from the SGZ and migrate to the GCL (*Kempermann, 2012*). After SE, there is a surge in proliferation in the SGZ, and neurons either migrate correctly to the GCL or aberrantly in the hilus (*Parent et al., 1997*).

These hilar ectopic GCs are thought to contribute to seizure generation in the epileptic brain because they are innervated by residual CA3 neurons, and project to GCs, making a major contribution to mossy fiber innervation of GCs in the inner molecular layer (*Scharfman et al., 2000*; *Kron et al., 2010*; *Pierce et al., 2011*; *Scharfman and Pierce, 2012*; *Althaus et al., 2016*). When epileptiform activity occurs in CA3 in slices of epileptic rats, CA3 evokes discharges in hilar ectopic GCs that in turn excite GCs in the GCL (*Scharfman et al., 2000*). Consistent with the idea that hilar ectopic GCs promote seizures, the numbers of hilar ectopic GCs are correlated with chronic seizure frequency in rats (*McCloskey et al., 2006*) and mice (*Hester and Danzer, 2013*). Furthermore, suppressing hilar ectopic GC formation reduces chronic seizures (*Jung et al., 2006*; *Cho et al., 2015*; *Hosford et al., 2016*).

Notably, this is the first study to our knowledge showing that increased hilar ectopic GCs were found in mice that had reduced seizures. One potential explanation is that SE-induced hippocampal damage was reduced in Cre+ females with high numbers of hilar ectopic GCs. Therefore, the circuitry of the DG would be very different compared to past studies of hilar ectopic GCs where a neuronal loss was severe . The presence of mossy cells is one way the circuitry would be different. MCs normally support the young adult-born GCs that migrate to the GCL (*Piatti and Schinder, 2018*). Mossy cells provide an important activator of newborn GCs when they are young (*Chancey et al., 2014*). Mossy cells also innervate hilar ectopic GCs (*Pierce et al., 2007*). Another possibility is that there was protection against chronic seizures in female Cre+ mice by increasing adult-born neurons *in the GCL*. The reason to suggest this possibility is that prior studies showed that young adult-born neurons in the GCL primarily inhibit GCs in the normal brain (*Drew et al., 2016*) and are relatively quiescent in the epileptic brain (*Jakubs et al., 2006*).

## Additional considerations

This study is limited by the possibilities of type II statistical errors in those instances where we divided groups by genotype and sex, leading to comparisons of 3–5 mice/group. Another potential caveat is that female mice were selected regardless of the stage of the estrous cycle.

## Conclusions

In the past, suppressing adult neurogenesis before SE was followed by fewer hilar ectopic GCs and reduced chronic seizures. Here, we show that the opposite - enhancing adult-born neurons before SE and increased hilar ectopic GCs - does not necessarily reduce seizures. We suggest instead that protection of the hilar neurons from SE-induced excitotoxicity was critical to reducing seizures. The reason for the suggestion is that the survival of hilar neurons would lead to the persistence of the normal inhibitory functions of hilar neurons, protecting against seizures. However, this is only a suggestion

at the present time because we do not have data to prove it. Additionally, because protection was in females, sex differences are likely to have played an important role. Regardless, the results show that enhancing-born neurons of young adult-born neurons in Nestin-Cre+ mice had a striking effect in the pilocarpine model, reducing chronic seizures in female mice.

# Materials and methods

## General information

Animal care and use was approved in Protocol #AP2016-557 by the Nathan Kline Institute Institutional Animal Care and Use Committee and met the regulations of the National Institute of Health and the New York State Department of Health. Mice were housed in standard mouse cages, with a 12 hr light/dark cycle and food (Laboratory rodent diet 5001; W.F. Fisher & Sons) and water ad libitum. During gestation and until weaning, mice were fed chow formulated for breeding (Formulab diet 5008; W.F. Fisher & Sons).

## Increasing adult-born neurons

To enhance-born neurons, a method was used that depends on deletion of *Bax*, the major regulator of programmed cell death in adult-born neurons (*Sun et al., 2004*; *Sahay et al., 2011b*; *Ikrar et al., 2013*; *Adlaf et al., 2017*). Enhancement of-born neurons was induced by conditional deletion of *Bax* from Nestin-expressing progenitors (*Sahay et al., 2011b*). These mice were created by crossing mice that have *loxP* sites flanking the pro-apoptotic gene *Bax* ($Bax^{fl/fl}$) with a Nestin-CreER$^{T2}$ mouse line in which tamoxifen-inducible Cre recombinase (CreER$^{T2}$) is expressed under the control of the rat *Nestin* promoter (*Sahay et al., 2011b*). It was shown that after tamoxifen injection in adult mice, there is an increase in dentate gyrus neurogenesis based on studies of bromo-deoxyuridine, Ki67, and double-cortin (*Sahay et al., 2011b*). The Nestin-CreER$^{T2}$$Bax^{fl/fl}$ mouse line was kindly provided by Drs. Amar Sahay and Rene Hen and used and described previously by our group (*Bermudez-Hernandez et al., 2017*; *Jain et al., 2019*). Although Nestin-Cre-ER$^{T2}$ mouse lines have been criticized because they can have leaky expression, the mouse line used in the present study did not (*Sun et al., 2014*), which we confirmed (*Jain et al., 2019*).

Starting at 6 weeks of age, mice were injected subcutaneously (s.c.) with tamoxifen (dose 100 mg/kg, 1 /day for 5 days; Cat# T5648, Sigma-Aldrich). Tamoxifen was administered from a stock solution (20 mg/ml in corn oil, containing 10% absolute alcohol; Cat# C8267, Sigma-Aldrich). Tamoxifen is light-sensitive so it was stored at 4 °C in an aluminum foil-wrapped container for the duration of treatment (5 days).

## Pilocarpine-induced SE

Six weeks after the last dose of tamoxifen injection, mice were injected with pilocarpine to induce SE. Methods were similar to those used previously (*Jain et al., 2019*). On the day of pilocarpine injection, there were 2 initial injections of pre-treatments and then one injection of pilocarpine. The first injection of pre-treatments was a solution of ethosuximide (150 mg/kg of 84 mg/ml in phosphate-buffered saline, s.c.; Cat# E;7138, Sigma-Aldrich). Ethosuximide was used because the background strain, C57BL6/J, is susceptible to respiratory arrest during a severe seizure and ethosuximide decreases the susceptibility (*Iyengar et al., 2015*). The second injection of pre-treatments was a solution of scopolamine methyl nitrate (1 mg/kg of 0.2 mg/ml in sterile 0.9% sodium chloride solution, s.c.; Cat# 2250, Sigma-Aldrich) and terbutaline hemisulfate (1 mg/kg of 0.2 mg/ml in sterile 0.9% sodium chloride solution, s.c.; Cat# T2528, Sigma-Aldrich). Scopolamine is a muscarinic cholinergic antagonist and when injected as methyl nitrate it does not cross the blood-brain barrier. Therefore, scopolamine decreased peripheral cholinergic side effects of the pilocarpine without interfering with the central actions of pilocarpine. Terbutaline was used to keep airways patent during severe seizures, minimizing mortality. Ethosuximide had to be administered separately because it precipitates when mixed with scopolamine and terbutaline.

Thirty minutes after the pre-treatments, pilocarpine hydrochloride was injected (260–280 mg/kg of 50 mg/ml in sterile 0.9% sodium chloride solution, s.c.; Cat# P6503; Sigma-Aldrich). Different doses were used because different batches of pilocarpine had different abilities to elicit SE.

The severity of SE was decreased by administering the benzodiazepine diazepam (10 mg/kg of 5 mg/ml stock solution, s.c.; NDC# 0409-3213-12, Hospira, Inc) 2 hr after pilocarpine injection. In females, diazepam was injected earlier, 40 min after the onset of the first seizure, because in the first group of females in which diazepam was injected 2 hr after pilocarpine, there was severe brain damage. While sedated with diazepam, animals were injected with warm (31 °C) lactated Ringer's solution (s.c.; NDC# 07-893-1389, Aspen Veterinary Resources). At the end of the day, mice were injected with ethosuximide using the same dose as before pilocarpine. For the next 3 days, chow was provided that was moistened with water. The cage was placed on a heating blanket to maintain the cage temperature at 31 °C.

## Stereotaxic surgery
### General information
Mice were anesthetized by isoflurane inhalation (3% isoflurane for induction and 1.75–2% isoflurane for maintenance during surgery; NDC# 07-893-1389, Patterson Veterinary) and placed in a stereotaxic apparatus (David Kopf Instruments). Prior to surgery, the analgesic Buprenex (Buprenorphine hydrochloride; NDC# 1296-0757-5; Reckitt Benckheiser) was diluted in sterile saline (0.9% sodium chloride solution) to yield a 0.03 mg/ml stock solution and 0.2 mg/kg was injected s.c. During surgery, mice were placed on a heating blanket with a rectal probe for automatic maintenance of body temperature at 31 °C.

### Implantation of EEG electrodes
Before electrode implantation, hair over the skull was shaved and then the scalp was cleaned with 70% ethanol. A midline incision was made to expose the skull with a sterile scalpel. To implant subdural screw electrodes (0.10' stainless steel screws, Cat# 8209, Pinnacle Technology), six holes were drilled over the exposed skull. The coordinates were: right occipital cortex (anterior-posterior or AP −3.5 mm from Bregma, mediolateral or ML, 2.0 mm from the midline); left frontal cortex (Lt FC, AP −0.5 mm; ML −1.5 mm); left hippocampus (AP −2.5 mm; ML −2.0 mm) and right hippocampus (AP −2.5 mm; ML 2.0 mm). An additional screw was placed over the right olfactory bulb as ground (AP 2.3 mm; ML 1.8 mm) and another screw over the cerebellum at the midline as reference (relative to Lambda: AP −1.5 mm; ML −0.5 mm). Here, 'ground' refers to the earth ground, and 'reference' refers to the reference for all 4 screw electrode recordings (*Moyer et al., 2017*). An eight-pin connector (Cat# ED85100-ND, Digi-Key Corporation) was placed over the skull and secured with dental cement (Cat# 51459, Dental Cement Kit; Stoelting Co.).

After surgery, mice were injected with 50 ml/kg warm (31 °C) lactated Ringer's solution (s.c.; NDC# 09355000476, Aspen Veterinary Resources). Mice were housed a clean cage on a heating blanket for 24 hr. Moistened food pellets were placed at the base of the cage to encourage food intake. Afterwards, mice were housed individually because group housing leads to disturbance of the implant by other mice in the cage.

## Continuous video-EEG recording and analysis
### Video-EEG recording
Mice were allowed 3 weeks to recover from surgery. During this time, mice were housed in the room where video-EEG equipment was placed so that mice could acclimate to the recording environment. To record video-EEG, the pin connector on the head of the mouse was attached to a preamplifier (Cat# 8406, Pinnacle Technology) which was attached to a commutator (Cat# 8408, Mouse Swivel/Commutator, four-channel, Pinnacle Technology) to allow freedom of movement. Signals were acquired at a 500 Hz sampling rate, and band-pass filtered at 1–100 Hz using Sirenia Acquisition software (https://www.pinnaclet.com, RRID:SCR_016183). Video was captured with a high-intensity infrared LED camera (Cat# PE-605EH, Pecham) and was synchronized to the EEG record.

To monitor pilocarpine-induced SE, video-EEG was recorded for an hour before ad 24 hr after the pilocarpine injection. Approximately 5–6 weeks after pilocarpine-induced SE, video-EEG was recorded to measure spontaneous recurrent seizures. Video-EEG was recorded continuously for 3 weeks.

## Video-EEG analysis

EEG was analyzed offline with Sirenia Seizure Pro, V2.0.7 (Pinnacle Technology, RRID:SCR_016184). A seizure was defined as a period of rhythmic (>3 Hz) deflections that were >2 x the standard deviation of the baseline mean and lasted at least 10 sec (*Jain et al., 2019*). Seizures were rated as convulsive if an electrographic seizure was accompanied by a behavioral convulsion (observed by video playback), defined as stages 3–5 using the Racine scale (*Racine, 1972*) where stage 3 is unilateral forelimb clonus, stage 4 is bilateral forelimb clonus with rearing, and stage 5 is bilateral forelimb clonus followed by rearing and falling. A seizure was defined as non-convulsive when there was electrographic evidence of a seizure but there were no stage 3–5 behavior.

SE was defined as continuous seizures for >5 min (*Chen and Wasterlain, 2006*) and EEG amplitude in all four channels >3 x the baseline mean. For mice without EEG, SE was defined by stage 3–5 seizures that did not stop with a resumption of normal behavior. Often stage 3–5 seizures heralded the onset of SE and then occurred intermittently for hours. In between convulsive behavior mice had twitching of their body, typically in a prone position.

SE duration was defined in light of the fact that the EEG did not return to normal after the initial period of intense activity. Instead, intermittent spiking occurred for at least 24 hr, as we previously described (*Jain et al., 2019*) and has been described by others (*Mazzuferi et al., 2012*; *Bumanglag and Sloviter, 2018*; *Smith et al., 2018*). We, therefore, chose a definition that captured the initial, intense activity. We defined the end of this time as the point when the amplitude of the EEG deflections were reduced to 50% or less of the peak deflections during the initial hour of SE. Specifically, we selected the time after the onset of SE when the EEG amplitude in at least three channels had dropped to approximately two times the amplitude of the EEG during the first hour of SE, and remained depressed for at least 10 min (Fig S2 in *Jain et al., 2019*). Thus, the duration of SE was defined as the time between the onset and this definition of the 'end' of SE.

To access the severity of chronic seizures, frequency and duration of seizures were measured during the 3 weeks of EEG recording. Inter-cluster interval was defined as the maximum number of days between two clusters.

## Tissue processing

### Perfusion-fixation and sectioning

Mice were perfused after video-EEG recording. To perfuse, mice were deeply anesthetized by isoflurane inhalation followed by urethane (250 mg/kg of 250 mg/ml in 0.9% sodium chloride, intraperitoneal, i.p.; Cat#U2500; Sigma-Aldrich). After loss of a reflex to a tail pinch, and loss of a righting reflex, consistent with deep anesthesia, the heart cavity was opened, and a 25-gauge needle was inserted into the heart, followed by perfusion with 10 ml saline 0.9% sodium chloride in double-distilled water (ddH$_2$O) using a peristaltic pump (Minipuls 1; Gilson) followed by 30 ml of cold (4 °C) 4% paraformaldehyde (PFA; Cat# 19210, Electron Microscopy Sciences) in 0.1 M phosphate buffer (PB; pH 7.4). The brains were removed immediately, hemisected, and post-fixed for at least 24 hr in 4% PFA at 4 °C. After post-fixation, one hemisphere was cut in the coronal plane and the other in the horizontal plane (50 μm-thick sections) using a vibratome (Cat# TPI-3000, Vibratome Co.). Sections were collected sequentially to select sections that were from similar septotemporal levels. For the dorsal hippocampus, coronal sections were selected every 300 μm starting at the first section where the DG blades are fully formed (between AP −1.94 and −2.06 mm). Horizontal sections were chosen every 300 μm starting from the temporal pole at the place where the GCL is clearly defined (between DV 0.84 and 1.08 mm). This scheme is diagrammed and described in more detail in prior studies (*Moretto et al., 2017*).

### Doublecortin

#### Procedures for staining

Doublecortin (DCX), a microtubule-associated protein (*Gleeson et al., 1999*), was used to identify immature adult-born neurons (*Brown et al., 2003*; *Couillard-Despres et al., 2005*), and was stained after antigen retrieval (*Botterill et al., 2015*). First, free-floating sections were washed in 0.1 M Tris buffer (TB, 3×5 min). Sections were then incubated in sodium citrate (Cat# S4641, Sigma-Aldrich) buffer (2.94 mg/ml in ddH$_2$O, pH 6.0 adjusted with HCl) in a preheated water bath at 85 °C for

30 min. Sections were washed with 0.1 M TB (3×5 min), blocked in 5% goat serum (Cat# S-1000, RRID:AB_2336615, Vector Laboratories) in 0.1 M TB with 0.5% (v/v) Triton X-100 (Cat# X-100, Sigma-Aldrich) and 1% (w/v) bovine serum albumin for 1 hr. Next, sections were incubated overnight with primary antibody (1:1000 diluted in blocking serum, monoclonal anti-rabbit DCX; Cat#4604 S, Cell Signaling Technology) on a shaker (Model# BDRAA115S, Stovall Life Science Inc) at room temperature.

On the next day, sections were washed in 0.1 M TB (3×5 min), treated with 2.5% hydrogen peroxide (Cat# 216763, Sigma-Aldrich) for 30 min to block endogenous peroxide, and washed with 0.1 M TB (3×5 min). Next, sections were incubated in secondary antibody (biotinylated goat anti-rabbit IgG, 1:500, Vector Laboratories) for 1 hr in 0.1 M TB, followed by washes with 0.1 M TB (3×5 min). Sections were then incubated in avidin-biotin complex (1:500 in 0.1 M Tris buffer; Cat# PK-6100, Vector) for 2 hr, washed in 0.1 M TB (1×5 min) and then in 0.175 M sodium acetate (14.36 mg/ml in ddH$_2$O, pH 6.8, adjusted with glacial acetic acid, 2×5 min; Cat# S8750, Sigma-Aldrich). Sections were reacted in 0.5 mg/ml 3, 3'-diaminobenzidine (DAB; Cat# D5905, Sigma-Aldrich) with 40 µg/ml ammonium chloride (Cat# A4514, Sigma-Aldrich), 3 µg/ml glucose oxidase (Cat# G2133, Sigma-Aldrich), 2 mg/ml (D+)-glucose (Cat# G5767, Sigma-Aldrich) and 25 mg/ml ammonium nickel sulfate (Cat# A1827, Sigma-Aldrich) in 0.175 M sodium acetate. Sections were washed in 0.175 M sodium acetate (2×5 min) and 0.1 M TB (5 min), mounted on gelatin-coated slides (1% bovine gelatin; Cat# G9391, Sigma-Aldrich), and dried overnight at room temperature.

On the next day, sections were dehydrated with increasing concentrations of ethanol, cleared in Xylene (Cat# 534–56, Sigma-Aldrich), and coverslipped with Permount (Cat# 17986–01, Electron Microscopy Sciences). Sections were viewed with a brightfield microscope (Model BX51; Olympus of America) and images were captured with a digital camera (Model Infinity3-6URC, Teledyne Lumenera).

## DCX analysis

DCX was quantified by first defining a region of interest (ROI) that included the adult-born cells and the majority of their DCX-labeled dendrites: the SGZ, GCL, and inner molecular layer. The SGZ was defined as a region that extended from the GCL into the hilus for a width of 100 µm because this region included the vast majority of the DCX immunoreactivity. The inner molecular layer was defined as the 100 µm immediately above the GCL. Next, a threshold was selected where DCX-immunoreactive (ir) cells were above, but the background was below the threshold, as described in more detail elsewhere (*Jain et al., 2019*).

This measurement is referred to as area fraction in the Results and expressed as a percent. For a given animal, the area fraction was determined for 3 coronal sections in the dorsal hippocampus between AP –1.94 to –2.06 mm and 3–4 horizontal sections in the ventral hippocampus between DV 0.84–1.08 mm, with sections spaced 300 µm apart. These area fractions were averaged so that a mean area fraction was defined for each animal. For these and other analyses described below, the investigator was blinded.

## Prox1

### Procedures for staining

In normal rodent adult brains, Prospero homeobox 1 (Prox1) is expressed in the GCs (*Pleasure et al., 2000*) and in the hilus (*Bermudez-Hernandez et al., 2017*). To stain for Prox1, free-floating sections were washed in 0.1 M TB pH 7.4, 3×5 min. Sections were then incubated in 0.1 M TB with 0.25%Triton X-100 for 30 min followed by a 10 min-long wash in 0.1 M TB with 0.1%Triton X-100 (referred to as Tris A). Next, sections were treated with 1% hydrogen peroxide in Tris A for 5 min followed by a 5 min-long wash in Tris A. Sections were blocked in 10% normal horse serum (Cat# S-2000, RRID:AB_2336617, Vector) in Tris A for 1 hr followed by a 10 min-long wash in Tris A and then 0.1 M TB with 0.1% Triton X-100 and 0.005% bovine serum albumin (referred to as Tris B). Next, sections were incubated overnight with primary antibody (goat anti-human Prox1 polyclonal antibody, 1:2000 diluted in Tris B, R, and D systems) rotated on a shaker (described above) at room temperature.

On the next day, sections were washed in Tris A and then in Tris B (5 min each). Sections were then incubated in secondary antibody (biotinylated anti-goat IgG made in horse, 1:500, Vector Laboratories, see Key Resources) for 1 hr in Tris B, followed by a wash with Tris A (5 min) and then Tris B (5 min), blocked in avidin-biotin complex (1:500 in Tris B) for 2 hr, and washed in 0.1 M TB (3×5 min). Sections were reacted in 0.5 mg/ml 3,3'-diaminobenzidine (DAB) with 40 µg/ml ammonium chloride, 3 µg/

ml glucose oxidase, 2 mg/ml (D+)-glucose and 5 mM nickel chloride (Cat# N6136, Sigma-Aldrich) in 0.1 M TB. Sections were washed in 0.1 M TB (3×5 min), mounted on 1% gelatin-coated slides and dried overnight at room temperature. On the next day, sections were dehydrated, cleared, and cover-slipped (as described above). Sections were viewed and images were captured as DCX above.

## Prox1 analysis

Prox1 was quantified in the hilus, defined based on zone 4 of *Amaral, 1978*. The definition of Amaral was modified to exclude 20 µm below the GCL (*Winawer et al., 2007*). The GCL boundary was defined as the location where GCs stopped being contiguous. Practically that meant there was no GC with more than a cell body width of cell-free space around it. A cell body width was 10 µm (*Claiborne et al., 1990*; *Amaral et al., 2007*).

CA3c was included in the ROI but hilar Prox1 cells have not been detected in the CA3c layer (*Scharfman et al., 2000*; *Winawer et al., 2007*). However, there are rare GCs in CA3 according to one study (*Szabadics et al., 2010*).

In ImageJ, a ROI was traced in the image taken at 20 x magnification and then a threshold was selected where Prox1-immunoreactivity was above the background threshold (*Jain et al., 2019*). Then Prox1 cells were counted using the Analyzed particle plugin where a particle with an area ≥10 µm$^2$ was counted. The following criteria were used to define a hilar Prox1-ir cell (*Bermudez-Hernandez et al., 2017*): (1) the hilar cell had sufficient Prox1-ir to reach a threshold equal to the average level of Prox1-ir of GCs in the adjacent GC layer, (2) All hilar Prox-ir cells were complete, i.e., not cut at the edge of the ROI. When hilar Prox1-ir cells were in clusters, although not many (typically 2–3 per 50 µm section), cells were counted manually. For each animal, three coronal sections in the dorsal hippocampus and 3–4 horizontal sections in the ventral hippocampus, with sections spaced 300 µm apart were chosen.

## Immunofluorescence

### Procedures for staining

Free-floating sections were washed (3×5 min) in 0.1 M TB followed by a 10 min long wash in Tris A and another 10 min-long wash in Tris B. Sections were incubated in blocking solution (5% normal goat serum or donkey serum in Tris B) for 1 hr at room temperature. Next, primary antibodies for anti-rabbit GluR2/3, anti-goat Prox1, anti-rabbit SOM, and anti-mouse parvalbumin (*Table 1*) were diluted in blocking solution and sections were incubated for 48 hr at 4 °C. For SOM labeling, antigen retrieval was used. Prior to the blocking step, sections were incubated in sodium citrate buffer (2.94 mg/ml in ddH$_2$O, pH 6.0 adjusted with HCl) in a preheated water bath at 85 °C for 30 min.

Next, sections were washed in Tris A and Tris B (10 min each) followed by 2 hr-long incubation in secondary antibody (1:500 in Tris B, see *Table 1*). Sections were washed in 0.1 M TB (3×5 min),

**Table 1.** Key resources.

| Primary antibodies | | | Secondary antibodies | | |
|---|---|---|---|---|---|
| Name | Dilution | Source, Identifier | Name | Dilution | Source, Identifier |
| anti-doublecortin (rabbit monoclonal) | 1:1000 | Cell Signaling Technology Cat# 4604 S, RRID:AB_10693771 | Biotinylated goat anti-rabbit IgG | 1:500 | Vector Laboratories Cat# BA-1000, RRID:AB_2313606 |
| anti-human Prox1 (goat polyclonal) | 1:2,000 | R and D systems Cat# AF2727, RRID:AB_2170716 | Biotinylated horse anti-goat IgG | 1:500 | Vector Laboratories Cat# BA-9500, RRID:AB_2336123 |
| anti-GluR2/3 (rabbit polyclonal) | 1:300 | Millipore Cat# AB1506, RRID:AB_90710 | Donkey anti-rabbit, Alexa Fluor 488 | 1:500 | Thermo Fisher Scientific Cat# A-21206, RRID:AB_2535792 |
| anti-Prox1 (goat polyclonal) | 1:2000 | R and D Systems Cat# AF2727, RRID:AB_2170716 | Donkey anti-goat, Alexa Fluor 546 | 1:500 | Thermo Fisher Scientific Cat# A-11056, RRID:AB_2534103 |
| anti-somatostatin (rabbit polyclonal) | 1:750 | Peninsula Laboratories Cat# T-4103.0050, RRID:AB_518614 | Goat anti-rabbit, Alexa Fluor 488 | 1:500 | Thermo Fisher Scientific Cat# A-11034, RRID:AB_2576217 |
| anti-parvalbumin (mouse monoclonal) | 1:1000 | Millipore Cat# MAB1572, RRID:AB_2174013 | Goat anti-mouse, Alexa Fluor 568 | 1:500 | Thermo Fisher Scientific Cat# A-11004, RRID:AB_2534072 |

and coverslipped with Citifluor AF1 mounting solution (Cat# 17970–25, Vector Labs). Images were captured on a confocal microscope (Model LSM 510 Meta; Carl Zeiss Microimaging).

## Procedures for analysis

GluR2/3-, SOM-, and parvalbumin- ir cells in the hilus and SGZ were counted from three dorsal and three ventral sections. Sections were viewed at 40 x of the confocal microscope for manual counts. Because ectopic GCs express GluR2/3, sections were co-labeled with Prox1. All co-labeled cells were considered as ectopic and excluded from the GluR2/3- ir cell counting to measure mossy cells.

## Fluorojade C

### Procedures for staining

Fluorojade C (FJ) is a fluorescent dye that is the 'gold standard' for stain degenerating neurons (*Schmued and Hopkins, 2000*; *Schmued et al., 2005*). First, sections were mounted on gelatin-coated slides (1% porcine gelatin in ddH2O; Cat# G1890, Sigma-Aldrich) and dried on a hot plate at 50–55°C for 1 hr. Then slides were placed in a staining rack and immersed in a 100% ethanol solution for 5 min, then in 70% ethanol for 2 min, followed by a 1 min wash in ddH2O.

Slides were then incubated in 0.06% potassium permanganate (Cat# P-279, Fisher Scientific) solution for 10 min on a shaker (described above) with gentle agitation, followed by washes in ddH2O (2×1 min). Slides were then incubated for 20 min in a 0.0002% solution of FJ in ddH2O with 0.1% acetic acid in the dark. The stock solution of FJ was 0.01% in ddH2O and was stored at 4 °C for up to 3 months. To prepare a working solution, 6 ml of stock solution was added to 294 mL of 0.1% acetic acid (Cat# UN2789, Fisher Scientific) in ddH2O and used within 10 min of preparation. Slides were subsequently protected from direct light. They were washed in ddH2O (3×1 min) and dried overnight at room temperature. On the next day, slides were cleared in Xylene (2×3 min) and coverslipped with DPX mounting medium (Cat# 44581, Sigma-Aldrich). Sections were photographed with an epifluorescence microscope (Model BX51; Olympus of America) and images were captured with a digital camera (Model Infinity3-6URC, Teledyne Lumenera).

### Procedures for analysis

We measured the FJ in the cell layers of CA1 and CA3. Manual counting of FJ-positive (FJ+) cells was not possible in these cell layers because there could be so many FJ+ cells that were overlapping. Instead, FJ staining in cell layers was quantified by first outlining the cell layer as a ROI at 10×magnification in ImageJ as before (*Jain et al., 2019*).

To outline the CA1 cell layer, the border with CA2 was defined as the point where the cell layer changed width, a sudden change that could be appreciated by the background in FJ-stained sections and confirmed by cresyl violet-stained sections. The border of CA1 and the subiculum was defined as the location where the normally compact CA1 cell layer suddenly dispersed. To outline CA3, the border with CA2 and CA3 was defined by the point where the stratum lucidum of CA3 terminated. This location was distinct in its background in FJ-stained sections. The border of CA3 and the hilus was defined according to zone 4 of *Amaral, 1978*. This location was also possible to detect in FJ-stained sections because the background in the hilus was relatively dark compared to area CA3.

After defining ROIs, a threshold fluorescence level was selected so that all cells that had very bright immunofluorescence were above the threshold but other cells that were similar in fluorescence to background staining were not (*Iyengar et al., 2015*; *Jain et al., 2019*). ImageJ was then used to calculate the area within the ROI and this measurement is referred to as the area fraction in the Results and expressed as a percent. For a given animal, the area fraction was determined for three coronal sections in the dorsal hippocampus between AP −1.94 and −2.06 mm and 3–4 horizontal sections in the ventral hippocampus between DV 0.84 and 1.08 mm, with sections spaced 300 µm apart. These area fractions were averaged so that a mean area fraction was defined for each animal.

## Data availability and statistical analyses

Data are presented as the mean ± standard error of the mean (SEM). All data are freely available at Open Science Framework. Digital object identifiers (DOIs) are: *Figure 1* and *Figure 1—figure supplements 1–4*, https://doi.org/10.17605/OSF.IO/VFTNG; *Figure 2* and *Figure 2—figure supplements 1 and 2*, https://doi.org/10.17605/OSF.IO/AK6U9; *Figure 3*, https://doi.org/10.17605/OSF.IO/2YZ97;

*Figure 4*, https://doi.org/10.17605/OSF.IO/TKEU5; *Figure 5*, https://doi.org/10.17605/OSF.IO/GYST4; *Figure 6*, https://doi.org/10.17605/OSF.IO/PCNGT; *Figure 7*, https://doi.org/10.17605/OSF.IO/2R5EJ; *Figure 8*, https://doi.org/10.17605/OSF.IO/7TJ2X. EEG and anatomical data were acquired and analyzed blinded to genotype and sex. Animals were assigned to groups in a random manner.

Statistical analyses were performed using GraphPad Prism Software (https://www.graphpad.com/scientific-software/prism/ RRID: SCR_002798). Statistical significance was set at $p < 0.05$. Robust regression and the Outlier removal (ROUT) method was used to remove outliers with ROUT coefficient Q set at 1%. Parametric tests were used when data fit a normal distribution, determined by the D'Agostino and Pearson or Shapiro-Wilk's normality tests, and there was homoscedasticity of variance (confirmed by an F-test). A Student's unpaired two-tailed t-test was used to assess differences between the two groups. A Welch's test was used instead of a Student's t-test when there was heteroscedasity of variance. One-way Analysis of Variance (ANOVA), two-way ANOVA, and three-way ANOVA were performed when there were multiple groups and were followed by Tukey's multiple comparison post-hoc test. The main factors for two-way ANOVA were genotype and sex; region was added as another factor for three-way ANOVA. Interaction between factors is reported in the Results if it was significant. A Fisher's exact test was used for comparing the proportions of binary data (yes/no). Pearson's Correlation was used to assess the association between two variables.

For data that did not follow a normal distribution, typically some data had a 0 value. In these cases, non-parametric tests were selected. The Mann-Whitney *U* test was used to compare two groups, and a Kruskal-Wallis test followed by post-hoc Dunn's test was used for multiple groups comparison.

## Acknowledgements

This study was supported by NIH R01 NS081203, NIH R37 NS126529, and NIH R01 AG055328 and the New York State Office of Mental Health.

## Additional information

### Competing interests

Helen E Scharfman: Reviewing editor, eLife. The other authors declare that no competing interests exist.

### Funding

| Funder | Grant reference number | Author |
| --- | --- | --- |
| National Institutes of Health | NS081203 | Helen E Scharfman |
| New York State Office of Mental Health | | John J LaFrancois Helen E Scharfman |
| National Institutes of Health | NS126529 | Helen E Scharfman |
| National Institutes of Health | AG055328 | Helen E Scharfman |

The funders had no role in study design, data collection and interpretation, or the decision to submit the work for publication.

### Author contributions

Swati Jain, Conceptualization, Investigation, Methodology, Writing – original draft, Writing – review and editing; John J LaFrancois, Kasey Gerencer, Meghan Kennedy, Methodology; Justin J Botterill, Supervision, Methodology, Writing – review and editing; Chiara Criscuolo, Formal analysis, Methodology, Writing – review and editing; Helen E Scharfman, Conceptualization, Data curation, Formal analysis, Supervision, Funding acquisition, Validation, Investigation, Methodology, Writing – original draft, Project administration, Writing – review and editing

## Author ORCIDs
Helen E Scharfman ⓘ https://orcid.org/0000-0003-4006-3383

## Ethics

This study was performed in strict accordance with the recommendations in the Guide for the Care and Use of Laboratory Animals of the National Institutes of Health. All of the animals were handled according to approved institutional animal care and use committee (IACUC) protocol (AP2016-557) of the Nathan Kline Institute. All surgery was performed under isoflurane anesthesia, and every effort was made to minimize suffering.

Reviewer #1 (Public Review): https://doi.org/10.7554/eLife.90893.3.sa1
Author response https://doi.org/10.7554/eLife.90893.3.sa2

---

## Additional files

### Supplementary files
• MDAR checklist

### Data availability

Data have been made available online at Open Science Framework (https://www.osf.org/).

The following datasets were generated:

| Author(s) | Year | Dataset title | Dataset URL | Database and Identifier |
|---|---|---|---|---|
| Scharfman H | 2024 | Data for Fig 1 | https://doi.org/10.17605/osf.io/vftng | Open Science Framework, 10.17605/osf.io/vftng |
| Scharfman H | 2024 | Data for Fig 2 | https://doi.org/10.17605/osf.io/ak6u9 | Open Science Framework, 10.17605/osf.io/ak6u9 |
| Scharfman H | 2024 | Data for Fig 3 | https://doi.org/10.17605/osf.io/2yz97 | Open Science Framework, 10.17605/osf.io/2yz97 |
| Scharfman H | 2024 | Data for Fig 4 | https://doi.org/10.17605/osf.io/tkeu5 | Open Science Framework, 10.17605/osf.io/tkeu5 |
| Scharfman H | 2024 | Data for Fig 5 | https://doi.org/10.17605/osf.io/gyst4 | Open Science Framework, 10.17605/osf.io/gyst4 |
| Scharfman H | 2024 | Data for Fig 6 | https://doi.org/10.17605/osf.io/pcngt | Open Science Framework, 10.17605/osf.io/pcngt |
| Scharfman H | 2024 | Data for Fig 7 | https://doi.org/10.17605/osf.io/2r5ej | Open Science Framework, 10.17605/osf.io/2r5ej |
| Scharfman H | 2024 | Data for Fig 8 | https://doi.org/10.17605/osf.io/7tj2x | Open Science Framework, 10.17605/osf.io/7tj2x |

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
