## [Editor Report · eLife assessment]

In this manuscript, Jain and colleagues explore whether increasing adult-born neurons is protective against status epilepticus and the development of spontaneous recurrent seizures (chronic epilepsy) in a mouse pilocarpine model of temporal lobe epilepsy. This is an **important** work that provides **solid** data, contradicting previous studies on suppressing chronic seizures by reduction in adult-born neurons.

---

## [Referee Report · Reviewer #1 (Public Review)]

Summary:

As adult-born granule neurons have been shown to play diverse roles, both positive and negative, to modulate hippocampal circuitry and function in epilepsy, understanding the mechanisms by which altered neurogenesis contribute to seizures is important for future therapeutic strategies. The work by Jain et al., demonstrates that increasing adult-born neurons (not increasing adult neurogenesis because BrdU birthdating was not performed in this study) before status epilepticus (SE) leads to a suppression in chronic seizures in the pilocarpine model of temporal lobe epilepsy. This work is potentially interesting because previous studies showed suppressing adult-born neurons led to reduced chronic seizures.

To increase adult-born neurons, the authors conditionally delete the pro-apoptotic gene Bax using a tamoxifen inducible Nestin-CreERT2 which has been previously published to increase proliferation and survival of adult-born neurons by Sahay et al. (although this was not shown in this study). After 6 weeks of tamoxifen injection, the authors subject male and female mice to pilocarpine induced SE. In the first study, at 2 hours after pilocarpine, the authors examine latency to the first seizure, severity and total number of acute seizures, and power during SE. In the second study in a separate group of mice, the authors examine chronic seizure number and frequency, seizure duration, postictal depression, and seizure distribution/cluster seizures for 3 weeks after pilocarpine. Overall, the study concludes that increasing adult-born neurons in the normal adult brain can reduce epilepsy in females specifically.

Strengths:

(1) The study is sex matched and reveals differences in response to increasing adult-born neurons in chronic seizures between male and females.

(2) The EEG recording parameters are stringent, and analysis of chronic seizures is comprehensive. In two separate experiments, the electrodes were implanted to record EEG from cortex as well as hippocampus. The recording is done for 10 hours post pilocarpine to analyze acute seizures, and for 3 weeks continuous video EEG recording was done to analyze chronic seizures.

Weaknesses:

(1) Increased DCX alone (without birthdating with BrdU) could indicate increased survival of adult-born neurons, not proliferation or birth of newborn neurons per se. While prior work has demonstrated that tamoxifen injection in adult mice showed an increase in dentate gyrus neurogenesis based on studies of BrdU, Ki67, and DCX (Sahay et al., 2011), the dynamics of adult-born neurons (proliferation, differentiation, and/or survival) could be different in epileptic (pilocarpine-treated) animals. Other stages, e.g., proliferation of neural precursors or maturation of adult-born dentate granule cells, was not examined. Analysis of additional stages of adult neurogenesis may reveal additional cellular understanding and add impact of the work on the field.

---

## [Author Response]

The following is the authors’ response to the original reviews.

**Public Reviews:**

**Reviewer #1 (Public Review):**
Summary:As adult-born granule neurons have been shown to play diverse roles, both positive and negative, to modulate hippocampal circuitry and function in epilepsy, understanding the mechanisms by which altered neurogenesis contributes to seizures is important for future therapeutic strategies. The work by Jain et al. demonstrates that increasing adult neurogenesis before status epilepticus (SE) leads to a suppression of chronic seizures in the pilocarpine model of temporal lobe epilepsy. This work is potentially interesting because previous studies showed suppressing neurogenesis led to reduced chronic seizures.To increase neurogenesis, the authors conditionally delete the pro-apoptotic gene Bax using a tamoxifen-inducible Nestin-CreERT2 which has been previously published to increase proliferation and survival of adult-born neurons by Sahay et al. After 6 weeks of tamoxifen injection, the authors subjected male and female mice to pilocarpine-induced SE. In the first study, at 2 hours after pilocarpine, the authors examine latency to the first seizure, severity and total number of acute seizures, and power during SE. In the second study in a separate group of mice, at 3 weeks after pilocarpine, the authors examine chronic seizure number and frequency, seizure duration, postictal depression, and seizure distribution/cluster seizures. Overall, the study concludes that increasing adult neurogenesis in the normal adult brain can reduce epilepsy in females specifically. However, important BrdU birthdating experiments in both male and female mice need to be included to support the conclusions made by the authors. Furthermore, speculative mechanisms lacking direct evidence reduce enthusiasm for the findings.

There are two suggestions. First, BrdU birthdating of newborn neurons is important to add to the paper so that there is support for the conclusions. Second, speculative text reduced enthusiasm. In response, we clarified the conclusions. We do not think that the clarified conclusions require BrdU birthdating (discussed further below). We also removed two schematics (and associated text) that we think the reviewer was referring to when speculation was mentioned.

We also want to point out something minor -that the times of injections listed above are not correct.

a. Seizures were not measured 2 hrs after pilocarpine; that is when the anticonvulsant diazepam was administered to males.

b. Seizures were not measured 3 weeks after pilocarpine; the duration of recording was 3 weeks.

(1) BrdU birthdating is required for conclusions.

We think that the Reviewer was suggesting birthdating because we were not clear about our conclusions, and we apologize for the confusion. The Reviewer stated that we concluded: “conditionally deleting Bax in Nestin-Cre+ cells leads to increased neurogenesis and hilar ectopic granule cells, thereby reducing chronic seizures.” (Note this is a quote from the review).

However, we did not intend to conclude that. We intended to conclude that conditionally deleting Bax in Nestin-Cre+ mice reduced chronic seizures in the mouse model of epilepsy that we used. Also, that conclusion only pertained to females. Please note we did not conclude that hilar ectopic granule cells led to reduced seizures. We also concluded that Bax deletion increased neurogenesis in female mice. We have revised the text to make the conclusions clear.

Abstract, starting on line 67:

The results suggest that selective Bax deletion to increase adult neurogenesis can reduce experimental epilepsy, and the effect shows a striking sex difference.

Results, starting on line 448:

Because Cre+ epileptic females had increased numbers of immature neurons relative to Cre- females at the time of SE, and prior studies show that Cre+ females had less neuronal damage after SE (Jain et al., 2019), female Cre+ mice might have had reduced chronic seizures because of high numbers of immature neurons. However, the data do not prove a causal role.

Starting on line 477:

...we hypothesized that female Cre+ mice would have fewer hilar ectopic GCs than female Cre- mice. However, that female Cre+ mice did not have fewer hilar ectopic GCs.

Discussion, starting on line 563:

The chronic seizures, measured 4-7 weeks after pilocarpine, were reduced in frequency by about 50% in females. Therefore, increasing young adult-born neurons before the epileptogenic insult can protect against epilepsy. However, we do not know if the protective effect was due to the greater number of new neurons before SE or other effects. Past data would suggest that increased numbers of newborn neurons before SE leads to a reduced SE duration and less neuronal damage in the days after SE. That would be likely to lessen the epilepsy after SE. However, there may have been additional effects of larger numbers of newborn neurons prior to SE.

Conclusions, starting on line 745:

In the past, suppressing adult neurogenesis before SE was followed by fewer hilar ectopic GCs and reduced chronic seizures. Here, we show that the opposite - enhancing adult neurogenesis before SE and increased hilar ectopic GCs - do not necessarily reduce seizures. We suggest instead that protection of the hilar neurons from SE-induced excitotoxicity was critical to reducing seizures. The reason for the suggestion is that the survival of hilar neurons would lead to persistence of the normal inhibitory functions of hilar neurons, protecting against seizures. However, this is only a suggestion at the present time because we do not have data to prove it. Additionally, because protection was in females, sex differences are likely to have played an important role. Regardless, the results show that enhancing neurogenesis of young adult-born neurons in Nestin-Cre+ mice had a striking effect in the pilocarpine model, reducing chronic seizures in female mice.

The Reviewer is correct that it would be interesting to know when the increase in adult neurogenesis occurred that was critical to the effect. For example, was it the initial increase following Bax deletion but before pilocarpine-induced SE, or the increase in neurogenesis following SE, or increased adult neurogenesis in the chronic stage of epilepsy. It also might be that related aspects of neurogenesis played a role such as the degree that maturation was normal in adult-born neurons. We have not pursued the experiments to identify these aspects of neurogenesis because of how much work it would entail. Also, approaches to conclude cause-effect relationships are going to be difficult.

(2) Speculation.

We removed the text and supplemental figures with schematics that we think were the overly speculative parts of the paper the Reviewer mentioned.

Strengths:(1) The study is sex-matched and reveals differences in response to increasing adult neurogenesis in chronic seizures between males and females.(2) The EEG recording parameters are stringent, and the analysis of chronic seizures is comprehensive. In two separate experiments, the electrodes were implanted to record EEG from the cortex as well as the hippocampus. The recording was done for 10 hours post pilocarpine to analyze acute seizures, and for 3 weeks continuous video EEG recording was done to analyze chronic seizures.Weaknesses:(1) Cells generated during acute seizures have different properties to cells generated in chronic seizures. In this study, the authors employ two bouts of neurogenesis stimuli (Bax deletion dependent and SE dependent), with two phases of epilepsy (acute and chronic). There are multiple confounding variables to effectively conclude that conditionally deleting Bax in Nestin-Cre+ cells leads to increased neurogenesis and hilar ectopic granule cells, thereby reducing chronic seizures.

As mentioned above, with a clarification of our conclusions we think we have addressed the concern. We believe that we conditionally deleted Bax in Nestin-expressing cells. We believe we found that female mice had reduced loss of hilar mossy cells and somatostatin-expressing neurons after SE, and fewer chronic seizures after SE. While it makes sense that increased neurogenesis caused the reduced seizures, we acknowledge it was not proved.

We do not make conclusions about the role of hilar ectopic granule cells. However, we note that they appear to have been similar in number across groups, which suggests they played no role in the results. This is very surprising and therefore adds novelty.

(2) Related to this is the degree of neurogenesis between Cre+ and Cre- mice and the nature of the sex differences. It is crucial to know the rate/fold change of increased neurogenesis before pilocarpine treatment and whether it is different between male and female mice.

We agree that if sex differences in adult neurogenesis could be shown by a sex difference in rate, fold change, maturation, and other characteristics. However, sex differences can also be shown by a change in doublecortin (DCX), which is what we did. We respectfully submit that we do not see an exhaustive study is critical.

As a result, we have clarified DCX was studied either before SE or in the period of chronic seizures:

Results, starting on line 406:

III. Before and after epileptogenesis, Cre+ female mice exhibited more immature neurons than Cre- female mice but that was not true for male mice.

Starting on line 446:

Therefore, elevated DCX occurred after chronic seizures had developed in Cre+ mice but the effect was limited to females.

Discussion, starting on line 592:

This study showed that conditional deletion of Bax from Nestin-expressing progenitors increased young adult-born neurons in the DG when studied 6 weeks after deletion and using DCX as a marker of immature neurons.

(3) The authors observe more hilar Prox1 cells in Cre+ mice compared to Cre- mice. The authors should confirm the source of the hilar Prox1+ cells.

This is an excellent question but it is unclear that it is critical to the seizures since both sexes showed more hilar Prox1 cells in Cre+ mice but only the females had fewer seizures than Cre- mice. This is the additional text to describe the results (starting on Line 493):

In past studies, hilar ectopic GCs have been suggested to promote seizures (Scharfman et al., 2000; Jung et al., 2006; Cho et al., 2015). Therefore, we asked if the numbers of hilar ectopic GCs correlated with the numbers of chronic seizures. When Cre- and Cre+ mice were compared (both sexes pooled), there was a correlation with numbers of chronic seizures (Fig. 6D1) but it suggested that more hilar ectopic GCs improved rather than worsened seizures. However, the correlation was only in Cre- mice, and when sexes were separated there was no correlation (Fig. 6D3).

When seizure-free interval was examined with sexes pooled, there was a correlation for Cre+ mice (Fig. 6D2) but not Cre- mice. Strangely, the correlations of Cre+ mice with seizure-free interval (Fig. 6D2, D4) suggest ectopic GCs shorten the seizure-free interval and therefore worsen epilepsy, opposite of the correlative data for numbers of chronic seizures. In light of these inconsistent results it seems that hilar ectopic granule cells had no consistent effect on chronic seizures.

(4) The biggest weakness is the lack of mechanism. The authors postulate a hypothetical mechanism to reconcile how increasing and decreasing adult-born neurons in GCL and hilus and loss of hilar mossy and SOM cells would lead to opposite effects - more or fewer seizures. The authors suggest the reason could be due to rewiring or no rewiring of hilar ectopic GCs, respectively, but do not provide clear-cut evidence.

As we mention above, we removed the supplemental figures with schematics because they probably were what seemed overly speculative.

We acknowledge that mechanism is not proven by our study. However, we would like to mention that in our view, showing preservation of hilar mossy cells and SOM cells, but not PV cells, does add mechanistic data to the paper. We understand more experiments are necessary.

**Reviewer #2 (Public Review):**
Summary:In this manuscript, Jain et al explore whether increasing adult neurogenesis is protective against status epilepticus (SE) and the development of spontaneous recurrent seizures (chronic epilepsy) in a mouse pilocarpine model of TLE. The authors increase adult neurogenesis via conditional deletion of Bax, a pro-apoptotic gene, in Nestin-CreERT2Baxfl/fl mice. Cre- littermates are used as controls for comparisons. In addition to characterizing seizure phenotypes, the authors also compare the abundance of hilar ectopic granule cells, mossy cells, hilar SOM interneurons, and the degree of neuronal damage between mice with increased neurogenesis (Cre+) vs Cre- controls. The authors find less severe SE and a reduction in chronic seizures in female mice with pre-insult increased adult-born neurons. Immunolabeling experiments show these females also have preservation of hilar mossy cells and somatostatin interneurons, suggesting the pre-insult increase in adult neurogenesis is protective.Strengths:(1) The finding that female mice with increased neurogenesis at the time of pilocarpine exposure have fewer seizures despite having increased hilar ectopic granule cells is very interesting.(2) The work builds nicely on the group's prior studies.(3) Apparent sex differences are a potentially important finding.(4) The immunohistochemistry data are compelling.(5) Good controls for EEG electrode implantation effects.(6) Nice analysis of most of the SE EEG data.Weaknesses:(1) In addition to the Cre- littermate controls, a no Tamoxifen treatment group is necessary to control for both insertional effects and leaky expression of the Nestin-CreERT2 transgene.

About “leaky” expression, we have not found expression to be leaky. We checked by injecting a Cre-dependent virus so that mCherry would be expressed in those cells that had Cre. The results were published as Supplemental Figure 9 in Jain et al. (2019).

In the revised manuscript we also mention a study that examined three Nestin-CreERT2 mouse lines (Sun et al., 2014). One of the mouse lines was ours. The leaky expression was not in the mouse line we use. We have added these points to the revised manuscript:

Methods, section II starting on line 791:

Although Nestin-Cre-ERT2 mouse lines have been criticized because they can have leaky expression, the mouse line used in the present study did not (Sun et al., 2014), which we confirmed (Jain et al., 2019).

(2) The authors suggest sex differences; however, experimental procedures differed between male and female mice (as the authors note). Female mice received diazepam 40 minutes after the first pilocarpine-induced seizure onset, whereas male mice did not receive diazepam until 2 hours post-onset. The former would likely lessen the effects of SE on the female mice. Therefore, sex differences cannot be accurately assessed by comparing these two groups, and instead, should be compared between mice with matching diazepam time courses.

We agree that a shorter delay between pilocarpine and diazepam would be likely to lead to less damage. However, the latency from pilocarpine to SE varied, making the time from the onset of SE to diazepam variable. Most of the variability was in females. By timing the diazepam injection differently in males and females, we could make the time from the onset of SE to diazepam similar between females and males. We had added a supplemental figure to show that our approach led to no significant differences between females and males in the latency to SE, time between SE and diazepam injection, and time between pilocarpine and diazepam injection. We also show that Cre+ females and Cre- females were not different in these times, so it could not be related to the neuroprotection of Cre+ females.

Additionally, the authors state that female mice that received diazepam 2 hours post-onset had severe brain damage. This is concerning as it would suggest that SE is more severe in the female than in the male mice.

We regret that our language was misleading. We intended to say females had more morbidity and mortality than males (lack of appetite and grooming, death in the days after SE) when we gave DZP 2 hrs after Pilo. We actually don’t know why because there were no differences in severity of SE. We think the females had worse outcome when they had a short latency to SE. These females had a longer period of SE before DZP than males, probably leading to worse outcome. To correct this we gave DZP to females sooner. Then morbidity and mortality was improved in females.

Interestingly, after we did this we saw females did not always have a short latency to SE. We maintained the same regimen however, to be consistent. As the new supplemental figure (above) shows, there were significant sex differences in the latency to SE, time between SE and DZP, and time between pilocarpine and DZP.

(3) Some sample sizes are low, particularly when sex and genotypes are split (n=3-5), which could cause a type II statistical error.

We agree and have noted this limitation in the Discussion:

Additional considerations, starting on line 739:

This study is limited by the possibilities of type II statistical errors in those instances where we divided groups by genotype and sex, leading to comparisons of 3-5 mice/group.

(4) Several figures show a datapoint in the sex and genotype-separated graphs that is missing from the corresponding male and female pooled graphs (Figs. 2C, 2D, 4B).

We are very grateful to the Reviewer for pointing out the errors. They are corrected.

(5) In Suppl Figs. 1B & 1C, subsections 1c and 2c, the EEG trace recording is described as the end of SE; however, SE appears to still be ongoing in these traces in the form of periodic discharges in the EEG.

The Reviewer is correct. It is a misconception that SE actually ends completely. The most intense seizure activity may, but what remains is abnormal activity that can last for days. Other investigators observe the same and have suggested that it argues against the concept of a silent period between SE and chronic epilepsy. We had discussed this in our prior papers and had referenced how we define SE. In the revised manuscript we add the information to the Methods section instead of referencing a prior study:

Methods, starting on line 899:

SE duration was defined in light of the fact that the EEG did not return to normal after the initial period of intense activity. Instead, intermittent spiking occurred for at least 24 hrs, as we previously described (Jain et al., 2019) and has been described by others (Mazzuferi et al., 2012; Bumanglag and Sloviter, 2018; Smith et al., 2018). We therefore chose a definition that captured the initial, intense activity. We defined the end of this time as the point when the amplitude of the EEG deflections were reduced to 50% or less of the peak deflections during the initial hour of SE. Specifically, we selected the time after the onset of SE when the EEG amplitude in at least 3 channels had dropped to approximately 2 times the amplitude of the EEG during the first hour of SE, and remained depressed for at least 10 min (Fig. S2 in Jain et al., 2019). Thus, the duration of SE was defined as the time between the onset and this definition of the "end" of SE.

(6) In Results section II.D and associated Fig.3, what the authors refer to as "postictal EEG depression" is more appropriately termed "postictal EEG suppression". Also, postictal EEG suppression has established criteria to define it that should be used.

We find suppression is typical in studies of ECT or humans (Esmaeili et al., 2023; Gascoigne et al., 2023; Hahn et al., 2023; Kavakbasi et al., 2023; Langroudi et al., 2023; Karl et al., 2024; Vilan et al., 2024; Zhao et al., 2024) and animal research uses the term postictal depression(Kanner et al., 2010; Krishnan and Bazhenov, 2011; Riljak et al., 2012; Singh et al., 2012; Carballosa-Gonzalez et al., 2013; Kommajosyula et al., 2016; Smith et al., 2018; Uva and de Curtis, 2020; Medvedeva et al., 2023). Therefore we think depression is a more suitable term.

The example traces in Fig. 3A and B should also be expanded to better show this potential phenomenon.

We expanded traces in Fig. 3 as suggested. They are in Fig 3A.

(7) In Fig.5D, the area fraction of DCX in Cre+ female mice is comparable to that of Cre- and Cre+ male mice. Is it possible that there is a ceiling effect in DCX expression that may explain why male Cre+ mice do not have a significant increase compared to male Cre- mice?

We thank the Reviewer for the intriguing possibility. We now mention it in the manuscript:

Results, starting on line 456:

It is notable that the Cre+ male mice did not show increased numbers of immature neurons at the time of chronic seizures but Cre+ females did. It is possible that there was a “ceiling” effect in DCX expression that would explain why male Cre+ mice did not have a significant increase in immature neurons relative to male Cre- mice.

(8) In Suppl. Fig 6, the authors should include DCX immunolabeling quantification from conditional Cre+ male mice used in this study, rather than showing data from a previous publication.

We have made this revision.

(9) In Fig 8, please also include Fluorojade-C staining and quantification for male mice.

The additional data for males have been added to part D.

(10) Page 13: Please specify in the first paragraph of the discussion that findings were specific to female mice with pre-insult increases in adult-born neurogenesis.

This has been done.

Minor:(11) In Fig. 1 and suppl. figure 1, please clarify whether traces are from male or female mice.

We have clarified.

(12) Please be consistent with indicating whether immunolabeling images are from female or male mice.a. Fig 5B images labeled as from "Cre- Females" and "Cre+ Females".b. Suppl. Fig 8: Images labeled as "Cre- F" and "Cre+ F".c. Fig 6: sex not specified.d. Fig. 7: sex only specified in the figure legend.e. Fig 8: only female mice were included in these experiments, but this is not clear from the figure title or legend.

We revised all figures according to the comments.

(13) Page 4: the last paragraph of the introduction belongs within the discussion section.

We recognize there is a classic view that any discussion of Results should not be in the Introduction. However, we find that view has faded and more authors make a brief summary statement about the Results at the end of the Introduction. We would like to do so because it allow Readers to understand the direction of the study at the outset, which we find is helpful.

(14) Page 6: The sentence "The data are consistent with prior studies..." is unnecessary.

We have removed the text.

(15) Suppl. Fig 6A: Please include representative images of normal condition DCX immunolabeling.

We have added these data. There is an image of a Cre- female, Cre+ female, Cre- male and Cre+ male in the new figure, Supplemental Figure 6. All mice had tamoxifen at 6 weeks of age and were perfused 6 weeks later. None of the mice had pilocarpine.

(16) In Suppl. Fig 7C, I believe the authors mean "no loss of hilar mossy and SOM cells" instead of "loss of hilar mossy and SOM cells".

This Figure was removed because of the input from Reviewer 1 suggesting it was too speculative.

**Reviewer #1 (Recommendations For The Authors):**
(1) The main claim of the study is that increasing adult neurogenesis decreases chronic seizures. However, to quantify adult-born neurons, DCX immunoreactivity is used as the sole metric to determine neurogenesis. This is insufficient as changes in DCX-expressing cells could also be an indicator of altered maturation, survival, and/or migration, not proliferation per se. To claim that increasing adult neurogenesis is associated with a reduction of chronic seizures, the authors should perform a pulse/chase (birth dating) experiment with BrdU and co-labeling with DCX.

We think that increased DCX does reflect increased adult neurogenesis. However, we agree that one does not know if it was due to increased proliferation, survival, etc. We also note that this mouse line has been studied thoroughly to show there was increased neurogenesis with BrdU, Ki67 and DCX. We mention that paper in the revised text:

Methods, starting on line 786:

It was shown that after tamoxifen injection in adult mice there is an increase in dentate gyrus neurogenesis based on studies of bromo-deoxyuridine, Ki67, and doublecortin (Sahay et al., 2011).

(2) As mentioned above, analysis of DCX staining alone months after TAM injections is limited. Instead, the cells could be labelled by BrdU prior to TAM injection, following which quantification of BrdU+/Prox1+ cells at 6 weeks post TAM injection should be performed in Cre+ and Cre- mice (males and females) to yield the rate of neurogenesis increase.

We respectfully disagree that birthdating cells is critical. Using DCX staining just before SE, we know the size of the population of cells that are immature at the time of SE. This is what we think is most important because these immature neurons are those that appear to affect SE, as we have already shown.

(3) To confirm the source of the hilar Prox1+ cells, a dual BrdU/EdU labeling approach would be beneficial. BrdU injection could be given before TAM injection and EdU injection before pilocarpine to label different cohorts of neural stem cells. Co-staining with Prox1 at different time points will help in identifying the origin of hilar ectopic cells.

We are grateful for the ideas of the Reviewer. We hesitate to do these experiments now because it seems like a new study to find out where hilar granule cells come from.

REFERENCES

Bumanglag AV, Sloviter RS (2018) No latency to dentate granule cell epileptogenesis in experimental temporal lobe epilepsy with hippocampal sclerosis. Epilepsia 59:2019-2034.

Carballosa-Gonzalez MM, Munoz LJ, Lopez-Alburquerque T, Pardal-Fernandez JM, Nava E, de Cabo C, Sancho C, Lopez DE (2013) EEG characterization of audiogenic seizures in the hamster strain gash:Sal. Epilepsy Res 106:318-325.

Cho KO, Lybrand ZR, Ito N, Brulet R, Tafacory F, Zhang L, Good L, Ure K, Kernie SG, Birnbaum SG, Scharfman HE, Eisch AJ, Hsieh J (2015) Aberrant hippocampal neurogenesis contributes to epilepsy and associated cognitive decline. Nat Commun 6:6606.

Esmaeili B, Weisholtz D, Tobochnik S, Dworetzky B, Friedman D, Kaffashi F, Cash S, Cha B, Laze J, Reich D, Farooque P, Gholipour T, Singleton M, Loparo K, Koubeissi M, Devinsky O, Lee JW (2023) Association between postictal EEG suppression, postictal autonomic dysfunction, and sudden unexpected death in epilepsy: Evidence from intracranial EEG. Clin Neurophysiol 146:109-117.

Gascoigne SJ, Waldmann L, Schroeder GM, Panagiotopoulou M, Blickwedel J, Chowdhury F, Cronie A, Diehl B, Duncan JS, Falconer J, Faulder R, Guan Y, Leach V, Livingstone S, Papasavvas C, Thomas RH, Wilson K, Taylor PN, Wang Y (2023) A library of quantitative markers of seizure severity. Epilepsia 64:1074-1086.

Hahn T et al. (2023) Towards a network control theory of electroconvulsive therapy response. PNAS Nexus 2:pgad032.

Jain S, LaFrancois JJ, Botterill JJ, Alcantara-Gonzalez D, Scharfman HE (2019) Adult neurogenesis in the mouse dentate gyrus protects the hippocampus from neuronal injury following severe seizures. Hippocampus 29:683-709.

Jung KH, Chu K, Lee ST, Kim J, Sinn DI, Kim JM, Park DK, Lee JJ, Kim SU, Kim M, Lee SK, Roh JK (2006) Cyclooxygenase-2 inhibitor, celecoxib, inhibits the altered hippocampal neurogenesis with attenuation of spontaneous recurrent seizures following pilocarpine-induced status epilepticus. Neurobiol Dis 23:237-246.

Kanner AM, Trimble M, Schmitz B (2010) Postictal affective episodes. Epilepsy Behav 19:156-158.

Karl S, Sartorius A, Aksay SS (2024) No effect of serum electrolyte levels on electroconvulsive therapy seizure quality parameters. J ECT 40:47-50.

Kavakbasi E, Stoelck A, Wagner NM, Baune BT (2023) Differences in cognitive adverse effects and seizure parameters between thiopental and propofol anesthesia for electroconvulsive therapy. J ECT 39:97-101.

Kommajosyula SP, Randall ME, Tupal S, Faingold CL (2016) Alcohol withdrawal in epileptic rats - effects on postictal depression, respiration, and death. Epilepsy Behav 64:9-14.

Krishnan GP, Bazhenov M (2011) Ionic dynamics mediate spontaneous termination of seizures and postictal depression state. J Neurosci 31:8870-8882.

Langroudi ME, Shams-Alizadeh N, Maroufi A, Rahmani K, Rahchamani M (2023) Association between postictal suppression and the therapeutic effects of electroconvulsive therapy: A systematic review. Asia Pac Psychiatry 15:e12544.

Mazzuferi M, Kumar G, Rospo C, Kaminski RM (2012) Rapid epileptogenesis in the mouse pilocarpine model: Video-EEG, pharmacokinetic and histopathological characterization. Exp Neurol 238:156-167.

Medvedeva TM, Sysoeva MV, Sysoev IV, Vinogradova LV (2023) Intracortical functional connectivity dynamics induced by reflex seizures. Exp Neurol 368:114480.

Riljak V, Maresova D, Jandova K, Bortelova J, Pokorny J (2012) Impact of chronic ethanol intake of rat mothers on the seizure susceptibility of their immature male offspring. Gen Physiol Biophys 31:173-177.

Sahay A, Scobie KN, Hill AS, O'Carroll CM, Kheirbek MA, Burghardt NS, Fenton AA, Dranovsky A, Hen R (2011) Increasing adult hippocampal neurogenesis is sufficient to improve pattern separation. Nature 472:466-470.

Scharfman HE, Goodman JH, Sollas AL (2000) Granule-like neurons at the hilar/CA3 border after status epilepticus and their synchrony with area CA3 pyramidal cells: Functional implications of seizure-induced neurogenesis. J Neurosci 20:6144-6158.

Singh B, Singh D, Goel RK (2012) Dual protective effect of passiflora incarnata in epilepsy and associated post-ictal depression. J Ethnopharmacol 139:273-279.

Smith ZZ, Benison AM, Bercum FM, Dudek FE, Barth DS (2018) Progression of convulsive and nonconvulsive seizures during epileptogenesis after pilocarpine-induced status epilepticus. J Neurophysiol 119:1818-1835.

Sun MY, Yetman MJ, Lee TC, Chen Y, Jankowsky JL (2014) Specificity and efficiency of reporter expression in adult neural progenitors vary substantially among nestin-creer(t2) lines. J Comp Neurol 522:1191-1208.

Uva L, de Curtis M (2020) Activity- and pH-dependent adenosine shifts at the end of a focal seizure in the entorhinal cortex. Epilepsy Res 165:106401.

Vilan A, Grangeia A, Ribeiro JM, Cilio MR, de Vries LS (2024) Distinctive amplitude-integrated EEG ictal pattern and targeted therapy with carbamazepine in KCNQ2 and KCNQ3 neonatal epilepsy: A case series. Neuropediatrics 55:32-41.

Zhao C, Tang Y, Xiao Y, Jiang P, Zhang Z, Gong Q, Zhou D (2024) Asymmetrical cortical surface area decrease in epilepsy patients with postictal generalized electroencephalography suppression. Cereb Cortex 34.